

# On the complex structure of Yang-Mills theory

Jan Horak[1], Jan M. Pawlowski[1,2] and Nicolas Wink[1,3]

**1** Institut für Theoretische Physik, Universität Heidelberg,
Philosophenweg 16, 69120 Heidelberg, Germany
**2** Institut für Kernphysik, Technische Universität Darmstadt,
Schlossgartenstraße 2, 64289 Darmstadt, Germany
**3** ExtreMe Matter Institute EMMI, GSI, Planckstr. 1, 64291 Darmstadt, Germany

## Abstract

We consider the coupled set of spectral Dyson-Schwinger equations in Yang-Mills theory for ghost and gluon propagators. Within this set-up, we perform a systematic analytic evaluation of the constraints on generalised spectral representations in Yang-Mills theory that are most relevant for informed spectral reconstructions. Furthermore, we provide numerical results for the coupled set of ghost and gluon spectral functions for a range of potential mass gaps of the gluon, while allowing for small violations of the spectral representation. The analyses are accompanied by thorough discussion of the limitations and extensions of the present work.

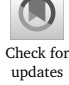

# 1  Introduction

The complete access to the hadronic bound state and resonance structure and to the non-perturbative dynamics of QCD at finite temperature and density requires the computation of timelike correlation functions. In functional approaches, such as the functional renormalisation group (fRG) or systems of Dyson-Schwinger–Bethe-Salpeter–Faddeev equations, the respective computations are carried out numerically, which requires a numerical non-perturbative approach to timelike correlation functions.

In [1] a *spectral* functional approach has been put forward, that utilises generalised spectral representations of correlation functions. This approach has been exemplified at the example of *spectral* DSEs in a $\phi^4$-theory. Moreover, consistent spectral renormalisation procedures have been constructed, which allow for a gauge-consistent regularisation in gauge theories. This framework has been used in [2] for the computation of the ghost spectral function in Yang-Mills theory. The latter result has been obtained from the spectral DSE of the ghost two-point function in Yang-Mills theory, using reconstruction results from [3] for the gluon spectral function as an input. Both works are based on the standard Källén-Lehmann (KL) representation of propagators or their dressing functions. In particular, [2] confirms the existence of the KL spectral representation for the ghost subject to one for the gluon, in the approximation used.

In the present work, we extend these analyses to the coupled set of spectral DSEs for the ghost and gluon propagators in Yang-Mills theory. Its numerical solution provides self-consistent results for ghost and gluon spectral functions, which we obtain while allowing for

small violations of the respective spectral representations. Apart from the numerical results, the spectral set-up allows us to unravel much of the intricate spectral structure of the Yang-Mills two-point functions in a fully analytic fashion. In summary, the present work serves a two-fold purpose: First, the results presented here constitute an important step towards full self-consistent functional resolution of timelike correlation functions which gives the access to the interesting scattering and resonance physics in QCD mentioned above. Second of all, both the numerical and the analytic results on the complex structure of ghost and gluon propagators provide non-trivial constraints for spectral reconstructions as well as direct computation of timelike propagators in Yang-Mills theory and QCD. Importantly, these constraints can be used to qualitatively improve the systematic error of these computations.

A first application of the latter structural results is the re-evaluation of the systematic error of existing computations. The results have the potential of significantly reducing the systematic uncertainties: In recent years, ghost and gluon spectral functions in Yang-Mills theory and QCD have been reconstructed from numerical data of Euclidean ghost and gluon propagators, see e.g. [3–8]. Direct computations have also been put forward, either perturbatively, e.g. [9, 10], with non-perturbative analytically continued DSEs [11, 12], or in a spectral approach [2, 13]. While these direct computations unravel interesting structures, they are still inconclusive.

We close the introduction with a bird's eye view on this work: In Section 2, we briefly review the basics of Yang-Mills theory and the spectral representations of gluon and ghost are discussed. In Section 3 we set up the coupled Yang-Mills system of gluon and ghost propagator DSEs in a spectral manner. Section 4 is devoted to a discussion of the complex structure of Yang-Mills theory based on the spectral formulation introduced in Section 3. In particular, we evaluate the non-spectral scenario of a pair of complex conjugate poles in the gluon propagator. In Section 5, we present numerical solutions to the coupled DSE system of Yang-Mills. Section 6 contains a conclusion and a discussion of the consequences of our combined results.

## 2 Yang-Mills theory and the spectral representation

We consider functional approaches to $3 + 1$-dimensional Yang-Mills theory with $N_c = 3$ colours in the Landau gauge, see [14–19] for fRG and [20–26] for DSE reviews. The gauge-fixed classical action is given as

$$S_{\text{YM}} = \int_x \left[ \frac{1}{4} F^a_{\mu\nu} F^a_{\mu\nu} - \bar{c}^a \partial_\mu D^{ab}_\mu c^b + \frac{1}{2\xi} (\partial_\mu A^a_\mu)^2 \right], \tag{1}$$

where $\xi$ denotes the gauge fixing parameter and $\int_x = \int \mathrm{d}^4 x$. Landau gauge is given in the limit $\xi \to 0$. Note that in (1) we have chosen a positive dispersion for the ghost. The field strength, $F_{\mu\nu}$, and covariant derivative $D_\mu$ in the adjoint representation, are given by

$$F^a_{\mu\nu} = \partial_\mu A^a_\nu - \partial_\nu A^a_\mu + g f^{abc} A^b_\mu A^c_\nu,$$

$$D^{ab}_\mu = \delta^{ab} \partial_\mu - g f^{abc} A^c_\mu, \tag{2}$$

with the usual structure constants $f^{abc}$ of SU(3).

The functional relations derived from (1) are one-loop exact in the fRG approach, and two-loop exact in the DSE approach, since the highest primitively divergent vertex is a four-point function. In both approaches the propagator plays a fundamental role,

$$\langle \phi_i(p) \phi_j(q) \rangle_c = (2\pi)^4 \delta(p + q) \mathcal{T}_{\phi_i \phi_j}(p) G_{\phi_i}(p), \tag{3}$$

where the subscript $_c$ stands for connected. The fields in (3) are $\phi = (A_\mu, c, \bar{c})$, and the tensor $\mathcal{T}_{\phi_i \phi_j}(p)$ carries the Lorenz and gauge group tensor structure. The scalar parts of the propagators are given by $G_\phi = G_A, G_c$. In the Landau gauge, the gluon propagator is transverse,

$$\mathcal{T}_{AA}(p)]_{\mu\nu}^{ab} = \delta^{ab} \Pi_{\mu\nu}^\perp(p), \qquad \Pi_{\mu\nu}^\perp(p) = \delta_{\mu\nu} - \frac{p_\mu p_\nu}{p^2}, \tag{4}$$

and $\Pi^\perp$ denotes the standard transverse projection operator. For the computations, we parametrise the scalar part $G_A$ of the gluon propagator as,

$$G_A(p) = \frac{1}{Z_A(p)\,p^2}, \tag{5}$$

where the gluon dressing function is given by $1/Z_A(p)$. Note that this convention might differ from other DSE related works and is more similar to fRG related conventions. Similarly, for the ghost we have a simple tensor structure $\mathcal{T}_{c\bar{c}}^{ab} = \delta^{ab}$, and we choose to parametrise the scalar part as

$$G_c(p) = \frac{1}{Z_c(p)\,p^2}, \tag{6}$$

with the ghost dressing function $1/Z_c(p)$. We will compute (6) for general complex momenta, of course including timelike ones. Extensions of correlation functions to the complex plane are particularly interesting, in view of their relevance for the self-consistent treatment of bound-state problems, see, e.g. [27–29].

If the KL spectral representation [30, 31] is applicable, a propagator $G$ can be recast in terms of its spectral function $\rho$,

$$G_\phi(p_0, |\vec{p}|) = \int_0^\infty \frac{d\lambda}{\pi} \frac{\lambda \rho_\phi(\lambda, |\vec{p}|)}{p_0^2 + \lambda^2}. \tag{7}$$

The spectral function naturally arises as the set of non-analyticities of the propagator in the complex momentum plane. If (7) holds, the non-analyticities are restricted to the real momentum axis. Equation (7) directly implies the following inverse relation between the spectral function and the retarded propagator,

$$\rho_\phi(\omega, |\vec{p}|) = 2 \operatorname{Im} G_\phi(-i(\omega + i0^+), |\vec{p}|), \tag{8}$$

where $\omega$ is a real frequency and $0^+$ implies that the limit is taken from above. Note also that Lorentz symmetry allows us to reduce our considerations to $\vec{p} = 0$ and then use $p_0^2 \to p^2$. Hence, for the remainder of this work, $|\vec{p}|$ will be dropped.

Formally, the ghost propagator is expected to obey the KL-representation [32, 33], if the corresponding propagator is causal. Also, recent reconstructions [6, 7] and calculations [2] show no signs of a violation of this property. The ghost spectral function must exhibit a single particle peak at vanishing spectral value, with residue $1/Z_c$. In addition, a continuous scattering tail is expected to show up in the spectral function via the logarithmic branch cut. This leads us to the general form of the ghost spectral function,

$$\rho_c(\omega) = \frac{\pi}{Z_c} \frac{\delta(\omega)}{\omega} + \tilde{\rho}_c(\omega), \tag{9}$$

where $\tilde{\rho}_c$ denotes the continuous tail of the spectral function and $\delta(\omega)/\omega$ has to be understood as a limiting process $\delta(\omega - 0^+)/\omega$.

Inserting (9) in (7) leads us to a spectral representation for the ghost dressing function,

$$\frac{1}{Z_c(p)} = \frac{1}{Z_c^0} + p^2 \int \frac{d\lambda}{\pi} \frac{\lambda \tilde{\rho}_c(\lambda)}{p^2 + \lambda^2}. \tag{10}$$

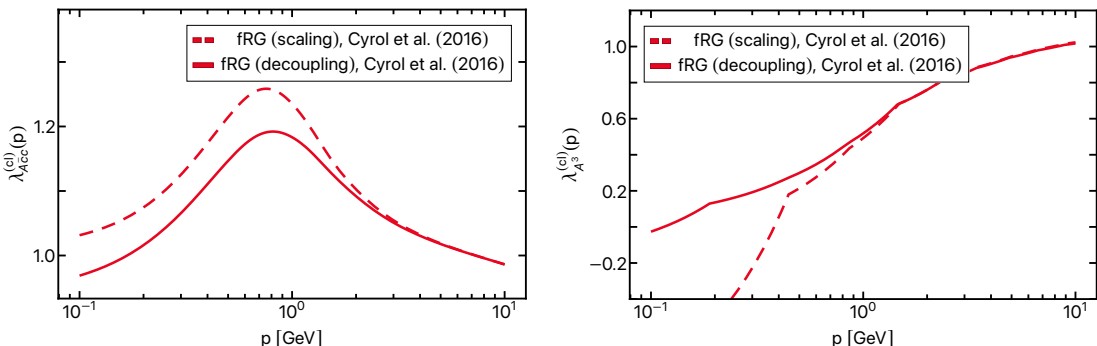

Figure 1: Ghost-gluon vertex dressing $\lambda_{Ac\bar{c}}^{(cl)}(p,q)$ (left) and three-gluon vertex dressing $\lambda_{A^3}^{(cl)}(p,q)$ (right), see (14) resp. (17). The data is taken from [3]. The dressing functions are shown at the symmetric point $p^2 = q^2 = (p+q)^2$ for scaling and lattice-type decoupling solution, more details can be found in [3].

It has been shown in [2] that the ghost spectral function obeys an analogue of the Oehme-Zimmermann superconvergence property of the gluon [34, 35]. Expressed in terms of the spectral representation of the dressing it reads

$$\int \frac{d\lambda}{\pi} \lambda \, \tilde{\rho}_c(\lambda) = -\frac{1}{Z_c^0} \,. \tag{11}$$

Equation (11) entails that the total spectral weight of the ghost vanishes. A generic discussion can be found in [2, 36].

The situation for the gluon is rather similar, as it has a spectral representation under the same conditions as the ghost, i.e. the propagator must be causal. In this case we are led to

$$G_A(p) = \int_0^\infty \frac{d\lambda}{\pi} \frac{\lambda \, \rho_A(\lambda)}{p^2 + \lambda^2} \,, \tag{12}$$

which is covered by (7). The associated sum rule is

$$\int_0^\infty \frac{d\lambda}{\pi} \lambda \, \rho_A(\lambda) = 0 \,, \tag{13}$$

the Oehme-Zimmermann superconvergence relation [34, 35]. In summary, both, ghost and gluon spectral function have a vanishing total spectral weight: (11) and (13). Note that the validity of the underlying assumptions is subject of an ongoing debate; for results and discussions, see, e.g. [2–4, 6, 10–12, 37–45].

Independent of this debate, the IR and UV of the gluon spectral function are fixed from analytic considerations, a detailed discussion thereof can be found in [3]. We briefly summarise it here: In Landau gauge, both, the IR and UV, the spectral function is negative. In the UV this simply follows from perturbation theory [34, 35]. For the IR, the situation is more intricate. In order to make statements, one requires that the gluon propagator is analytic in the finite, open semicircle in the upper half plane around the origin. This includes the Euclidean domain, and e.g. (7) meets this criterium. With this at hand, it can be shown that the gluon spectral function is negative in the IR, owing to the contribution of the ghost loop. More details of the derivation and explicit analytic forms can be found in [3].

# 3 Spectral DSEs of Yang-Mills theory

In this section, we set up the spectral Yang-Mills system in order to compute the gluon and ghost spectral function $\rho_A$ and $\rho_c$.

## 3.1 Vertex approximation

The full ghost-gluon vertex consists of two tensor structures, see e.g. [46–48],

$$[\Gamma_{A\bar{c}c}]_\mu^{abc}(p,q) = \mathrm{i} f^{abc}[q_\mu \lambda_{A\bar{c}c}^{(\mathrm{cl})}(p,q) + p_\mu \lambda_{A\bar{c}c}^{(\mathrm{nc})}(p,q)], \tag{14}$$

and the momentum arguments $p_i$ in our vertices $\Gamma_{\phi_1\cdots\phi_n}(p_1,...,p_{n-1})$ always indicate the incoming momentum of the field $\phi_i$. In (14) we have the incoming gluon momentum $p$ and anti-ghost momentum $q$, and we have dropped the momentum conserving $\delta$-function.

The ghost-gluon vertex is subject to Taylor's non-renormalisation theorem, and does not require renormalisation in the Landau gauge. Within our MOM-type scheme, the dressing functions are set to unity at the renormalisation point $\mu_{\mathrm{RG}}$, i.e., $Z_A(\mu_{\mathrm{RG}}) = Z_c(\mu_{\mathrm{RG}}) = 1$. Accordingly, the classical ghost-gluon dressing reduces to the strong coupling $g_s$ at the renormalisation point, which typically is chosen to be the symmetric point, $p^2 = q^2 = (p+q)^2$, or the soft gluon limit, $p \to 0, q^2 = \mu^2$. In short,

$$\lambda_{A\bar{c}c}^{(\mathrm{cl})}(p,q)\Big|_{\mu_{\mathrm{RG}}} = g_s. \tag{15}$$

We emphasise that (15) is not an RG-condition, it is a consequence of the non-renormalisation of the ghost-gluon vertex. Moreover, the non-classical dressing in (14) is proportional to the gluon momentum and hence drops out of the ghost DSE due to the transversality of the Landau gauge gluon propagator.

The lack of a logarithmic RG running also leads to a very mild momentum dependence of the vertex, see e.g. [46, 48–56]. In the left panel of Figure 1 the ghost-gluon vertex data from [46] is depicted at the symmetric point $p^2 = q^2 = (p+q)^2$ for both, the scaling solution and a lattice-type decoupling solution. For further explanations, we refer to the detailed discussion of [46, 47].

In the present work we neglect the mild momentum dependence and identify the vertex dressing $\lambda_{A\bar{c}c}^{(\mathrm{cl})}$ with its value at the renormalisation point, (14), to wit,

$$\lambda_{A\bar{c}c}^{(\mathrm{cl})}(p,q) \approx g_s, \tag{16}$$

which should only introduce a small systematic error for our Euclidean results.

The three-gluon vertex can be parametrised by ten longitudinal and four transverse tensor structures. In the Landau gauge, only the transverse ones contribute, the dominant being the classical tensor structure [57]. Neglecting the subleading tensor structures, the three-gluon vertex can be written as

$$[\Gamma_{A^3}^{(3)}]_{\mu\nu\rho}^{abc}(p,q) = \mathrm{i} f^{abc} \lambda_{A^3}^{(\mathrm{cl})}(p,q)[\mathcal{T}_{A^3}^{(\mathrm{cl})}]_{\mu\nu\rho}(p,q), \tag{17}$$

with the classical Lorentz structure $\mathcal{T}_{A^3}^{(\mathrm{cl})}$ defined as

$$[\mathcal{T}_{A^3}^{(\mathrm{cl})}]_{\mu\nu\rho}(p,q) = (p-q)_\nu \delta_{\mu\rho} + (2q+p)_\mu \delta_{\nu\rho} - (2p+q)_\rho \delta_{\mu\nu}. \tag{18}$$

At the symmetric momentum configuration, the dressing function $\lambda_{A^3}^{(\mathrm{cl})}$ gets negative in the deep IR region and rising for increasing momenta [46, 55, 58, 59] due to its anomalous dimension, see right panel of Figure 1. Since the ghost loop is known to dominate the gluon gap

equation in the IR, we approximate the dressing function by its counterpart at the renormalisation point, as already done for the ghost-gluon vertex, (16),

$$\lambda_{A^3}^{(cl)}(p,q) \approx g_s \, , \tag{19}$$

with $g_s$ being the strong running coupling $g_s(p)$ at the renormalisation scale $p^2 = \mu^2$. This yields a considerable technical simplification, since the real-time nature of the spectral approach requires all momentum integrals to be solved analytically, as discussed in detail in [2]. However, in contradistinction to the approximation in the ghost-gluon vertex this introduces a sizeable systematic error due to the sizeable momentum dependence shown in the right panel of Figure 1. Accordingly, we expect our results to be of qualitative nature, and the systematic error can be evaluated by comparing the results to those obtained in quantitatively reliable approximations within functional approaches, e.g. [46,55] and on the lattice, see e.g. [60–62].

We emphasise that our approach is by no means restricted to classical vertices: Quantum corrections may be duly accounted for, as long as the momentum loops involved can be integrated analytically. Especially, upon construction of spectral representations for higher $n$-point-functions, see e.g., [63–75], fully dressed vertices of general form can be included. In the present work, we restrict ourselves to classical ones, as this allows us to study the emergence and interrelations of poles and generic complex structures of the propagators themselves.

## 3.2 Spectral DSEs

In the Landau gauge, functional relations of transverse correlation functions are closed: They do not depend on the longitudinal sector due to the transversality of the gluon propagator, see [19, 46, 76]. For the present coupled set of propagator DSEs this entails that the gluon two-point function, $\Gamma_{AA}^{(2),\|}$, does not enter in the system: Neither the loop in the ghost DSE nor those in the gluon DSE depend on it. The ghost and gluon gap equations can be reduced to DSEs of the respective scalar parts, and we use the parametrisation,

$$[\Gamma_{AA}^{(2),\perp}]_{\mu\nu}^{ab}(p) = \Pi_{\mu\nu}^{\perp}(p)\delta^{ab}Z_A(p)p^2 \, , \tag{20}$$

$$[\Gamma_{c\bar{c}}^{(2)}]^{ab}(p) = \delta^{ab}Z_c(p)p^2 \, .$$

The dressings $Z_\phi(p)$ in (20) can be conveniently written in terms of the respective self energies, to wit,

$$Z_A(p)p^2 = Z_3 p^2 - \Sigma_{AA}(p) \, , \tag{21}$$

$$Z_c(p)p^2 = \tilde{Z}_3 p^2 - \Sigma_{\bar{c}c}(p) \, ,$$

with the renormalisation constants $Z_3$ and $\tilde{Z}_3$ associated with the gluon and ghost fields. They contain the counter terms, that lead to finite loops as well as adjusting the renormalisation conditions in their respective DSEs.

Similarly, the classical ghost-gluon and three-gluon vertices in Figure 2 contain the respective renormalisation constants $Z_1$ and $\tilde{Z}_1$, i.e.

$$S_{Ac\bar{c}}^{\mu}(p,q) = -\tilde{Z}_1 \mathrm{i}g_s f^{abc} p^\mu \, , \tag{22}$$

$$S_{A^3}^{\mu\nu\rho}(p,q) = Z_1 \mathrm{i}g_s f^{abc}[\mathcal{T}_{A^3}^{(cl)}]^{\mu\nu\rho}(p,q) \, .$$

As for the propagators, they contain the counterterms leading to finite loops and adjusting the renormalisation conditions in their respective DSEs. However, the ghost-gluon vertex does not require renormalisation in the Landau gauge, and we do not consider vertex DSEs. Accordingly, their consistent choice is $Z_1 = \tilde{Z}_1 = 1$, which is implemented later. For the time

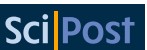

Figure 2: Diagrammatic representation of the Dyson-Schwinger equations for of the inverse gluon and ghost propagator. Blue dots represent full 1-PI vertices. Internal lines stand for full propagators.

being, we keep the renormalisation constants as they elucidate the systematics of the spectral renormalisation applied in Section 3.3.

The gluon and ghost self-energies $\Sigma_{AA}$ and $\Sigma_{\bar{c}c}$ in (21) contain all quantum corrections of the two-point functions, and are determined via their respective propagator DSEs. While the ghost DSE is one loop closed, the gluon DSE is two-loop closed, and we have dropped the two-loop diagrams. The corresponding system of DSEs for gluon and ghost two-point functions in (20) is depicted in Figure 2, with the notation as defined in Figure 3. The self-energies are then just given by the sum of all loop diagrams. We recast the gluon self-energy defined in (21) in terms of its two contributing one-loop diagrams as

$$\Sigma_{AA}(p) = \frac{1}{2}\Big(\mathcal{D}_{\text{gluon}}(p) - \mathcal{D}_{\text{ghost}}(p)\Big), \tag{23}$$

where $\mathcal{D}_{\text{gluon}}$ represents the gluon and $\mathcal{D}_{\text{ghost}}$ the ghost loop. With the classical vertex approximation discussed in Section 3.1, we arrive at

$$\mathcal{D}_{\text{gluon}} = g^2 C_A Z_1 \Pi^\perp_{\mu\nu}(p) \int_q G_A(p+q)\Pi^\perp_{\gamma\delta}(p+q)G_A(q)\Pi^\perp_{\alpha\beta}(q)[\mathcal{T}_{A^3}]_{\mu\alpha\gamma}(p,q)[\mathcal{T}_{A^3}]_{\delta\beta\nu}(-q,-p),$$

$$\mathcal{D}_{\text{ghost}} = g^2 C_A \tilde{Z}_1 \Pi^\perp_{\mu\nu}(p) \int_q G_c(p)G_c(p+q)q_\nu(p+q)_\mu. \tag{24}$$

Here, $C_A = N_c$ is the second Casimir for SU($N_c$) in the adjoint representation.

The ghost-self energy (21) reads,

$$\Sigma_{\bar{c}c}(p) = g^2 C_A \tilde{Z}_1 \int_q \Big(p^2 - \frac{(p \cdot q)^2}{q^2}\Big) G_A(q)G_c(p+q). \tag{25}$$

Now we recast the diagrams in (23) and (25) in their spectral form, using the KL representation (7) for both gluon and ghost propagators and contracting the Lorentz structure in (24) and (25). This leads us to

$$\mathcal{D}_{\text{gluon}} = g^2 N_c Z_1 \int_{\lambda_1,\lambda_2} \rho_A(\lambda_1)\rho_A(\lambda_2) \int_q V(p,q)\frac{1}{q^2+\lambda_1^2}\frac{1}{(p+q)^2+\lambda_2^2}, \tag{26a}$$

with

$$V(p,q) = \Pi^\perp_{\mu\nu}(p)\Pi^\perp_{\gamma\delta}(p+q)\Pi^\perp_{\alpha\beta}(q)[\mathcal{T}_{A^3}]_{\mu\alpha\gamma}(p,q)[\mathcal{T}_{A^3}]_{\delta\beta\nu}(-q,-p), \tag{26b}$$

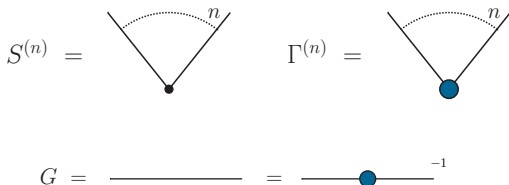

Figure 3: Diagrammatic notation used throughout this work: Lines stand for full propagators, small black dots stand for classical vertices, and larger blue dots stand for full vertices.

for the gluonic diagram in the gluon DSE. The function $V$ defined in (26b) captures all momentum dependencies arising from contracting the Lorentz structure of vertices and projection operators in (24).

The ghost diagram is given by

$$\mathcal{D}_{\text{ghost}} = g^2 N_c \tilde{Z}_1 \int_{\lambda_1, \lambda_2} \rho_c(\lambda_1) \rho_c(\lambda_2) \int_q \left( q^2 - \frac{(p \cdot q)^2}{p^2} \right) \frac{1}{q^2 + \lambda_1^2} \frac{1}{(p+q)^2 + \lambda_2^2} . \tag{26c}$$

Finally, the spectral representation of the ghost DSE reads,

$$\Sigma_{\bar{c}c}(p) = g^2 N_c \tilde{Z}_1 \int_{\lambda_1, \lambda_2} \rho_A(\lambda_1) \rho_c(\lambda_2) \int_q \left( p^2 - \frac{(p \cdot q)^2}{q^2} \right) \frac{1}{q^2 + \lambda_1^2} \frac{1}{(p+q)^2 + \lambda_2^2} , \tag{26d}$$

with $\rho_A$ and $\rho_c$ the gluon and ghost spectral functions, respectively, and $\int_\lambda := \int_0^\infty \mathrm{d}\lambda\, \lambda/\pi$. The momentum integrals are regularised with dimensional regularisation. Importantly, this makes both, the momentum and spectral integrations, finite, and allows us to interchange the order of spectral and momentum integration, as done in (26).

### 3.3 Spectral renormalisation

The momentum integrals in (26) involve two classical propagators with spectral masses $\lambda_1$ and $\lambda_2$ These are readily computed in $d = 4 - 2\epsilon$ dimensions; for the computational details and the final expressions see Appendix B. This leaves us with two spectral integrals.

Naively one could try to resort to a momentum space subtraction scheme by simply dropping the $1/\epsilon$-term arising from the momentum integration. However, the spectral integrals suffer from the same superficial degree of divergence as their respective momentum integral, and this naive implementation of a MOM scheme does not work. This is a generic feature in the spectral DSE, for a thorough discussion see [1]. There we have set up two spectral renormalisation schemes: *spectral dimensional renormalisation* and *spectral BPHZ renormalisation*, both exploiting the advantageous properties of dimensional regularisation of the momentum loop, but treating the spectral divergences differently.

Spectral dimensional renormalisation also treats the spectral integrals in dimensional regularisation, hence manifestly respecting all internal symmetries of the theory, including gauge theory. This property entails that the gluon gap equation in Yang-Mills theory has no quadratic divergence in spectral dimension renormalisation, and only logarithmic divergences related to the gluon wave function renormalisation are present.

In turn, in spectral BPHZ renormalisation quadratic divergences are present, which is a well-known property of the BPHZ scheme in gauge theories and originates in it being a momentum cutoff scheme. For a detailed discussion see [14,46,77,78] where also the direct link

to Wilsonian cutoffs in the fRG approach and the ensuing modified Slavnov-Taylor identities (STIs) is discussed. In short, momentum cutoff schemes such as BPHZ-type schemes necessitate a gluon mass counterterm, which is adjusted such that the STIs are satisfied. Accordingly, the occurrence of mass counterterms in Yang-Mills theory in a BPHZ-type scheme is a property of the scheme and *restores* gauge consistency and does not (necessarily) signal its breaking.

In the present spectral BPHZ set-up, the spectral divergences are cured by introducing counterterms, including a gluon mass counterterm, through the renormalisation constants in (21) and taking $\epsilon \to 0$ before computing the spectral integrals. Then, gauge invariance is restored by adjusting the finite part of this counterterm such that the STIs are satisfied at the level of the renormalised correlation function. For discussions about the treatment of quadratic divergences in functional approaches to Yang-Mills theory, see e.g. [46, 47, 55].

In summary, this amounts to a modification of the gluon DSE in (21) according to

$$Z_A(p)p^2 = Z_3 p^2 + \tilde{m}_A^2 - \Sigma_{AA}(p), \tag{27}$$

where the mass counterterm $\tilde{m}_A^2$ is chosen such that the quadratic divergence in $\Sigma_{AA}$ is cancelled. This already effectively absorbs the tadpole in the gluon DSE into the mass counterterm.

The ghost self-energy $\Sigma_{\bar{c}c}$ only carries a logarithmic divergence proportional to $p^2$, which can be subtracted by a proper choice of $\tilde{Z}_3$. Within spectral BPHZ renormalisation, the counterterms are chosen to be proportional to the respective self-energies $\Sigma$, evaluated at some RG scale $\mu_{\text{RG}}$. We use standard renormalisation condition for the (inverse) dressing functions,

$$Z_A(\mu_{\text{RG}}) = 1 + \frac{m_A^2}{\mu_{\text{RG}}^2}, \tag{28a}$$

$$Z_c(\mu_{\text{RG}}) = 1.$$

These renormalisation conditions are implemented by the respective choice of the renormalisation constants $Z_3, \tilde{m}_A^2$ and $\tilde{Z}_3$ as

$$Z_3 = 1 + \frac{\Sigma_{AA}(\mu_{\text{RG}})}{\mu_{\text{RG}}^2}, \tag{28b}$$

$$\tilde{m}_A^2 = m_A^2 + \Sigma_{AA}(\mu_{\text{RG}}),$$

$$\tilde{Z}_3 = 1 + \frac{\Sigma_{\bar{c}c}(\mu_{\text{RG}})}{\mu_{\text{RG}}^2},$$

augmented with $Z_1, \tilde{Z}_1 \to 1$, reflecting the lack of vertex DSEs. For a detailed discussion of self-consistent MOM-type RG conditions for DSEs (MOM in DSEs and MOM$^2$ in fRG equations and DSEs), see [79]. Eventually, this leads us to the renormalised system of DSEs for the gluon and ghost dressing functions,

$$Z_A(p)p^2 = p^2 + m_A^2 - \left[ \Sigma_{AA}(p) - \Sigma_{AA}(\mu_{\text{RG}}) \left( 1 + \frac{p^2 - \mu_{\text{RG}}^2}{\mu_{\text{RG}}^2} \right) \right],$$

$$Z_c(p)p^2 = p^2 - \left[ \Sigma_{\bar{c}c}(p) - \frac{p^2}{\mu_{\text{RG}}^2} \Sigma_{\bar{c}c}(\mu_{\text{RG}}) \right]. \tag{29}$$

In perturbative applications the mass parameter $m_A^2$ is chosen such that the gluon two point function has no infrared mass, tantamount to $Z_A(p)p^2 \to 0$ for $p \to 0$. This is the requirement of perturbative BRST symmetry, implying the equivalence of the transverse mass and the longitudinal one, and the latter vanishes due to the STI. In (29) this amounts to

$$m_A^2 = \Sigma_{AA}(0), \tag{30}$$

which reinstates perturbative gauge consistency with a massless gluon within the BPHZ-scheme.

In the IR, $m_A^2$ is linked to the dynamical emergence of the gluon mass gap in QCD, and its explicit choice of $m_A^2$ will be discussed in Section 5.

### 3.4 Evaluation at real frequencies

Apart from the integration over real spectral parameters $\lambda$, the renormalised DSEs in (29) can be evaluated *analytically* for general complex frequencies. For the extraction of the spectral functions with (8) we choose $p_0 = -i(\omega + i0^+)$. This leads us to the Minkowski variant of (29),

$$Z_A(\omega)\omega^2 = \omega^2 - m_A^2 + \left[ \Sigma_{AA}(\omega) - \Sigma_{AA}(\mu_{\mathrm{RG}}) \left( 1 - \frac{\omega^2 + \mu_{\mathrm{RG}}^2}{\mu_{\mathrm{RG}}^2} \right) \right], \tag{31a}$$

$$Z_c(p)\omega^2 = \omega^2 + \left[ \Sigma_{\bar{c}c}(\omega) + \frac{\omega^2}{\mu_{\mathrm{RG}}^2} \Sigma_{\bar{c}c}(\mu_{\mathrm{RG}}) \right], \tag{31b}$$

where, in a slight abuse of notation, we define $\Sigma(\omega) = \Sigma(-i(\omega + i0^+))$.

The explicit spectral integral expressions for the self-energies and their renormalised counterparts can be found in Appendix B. The remaining finite spectral integrals have to be computed numerically, and the spectral functions $\rho_{c,A}(\omega)$ are given with (8) as

$$\rho_A(\omega) = -\frac{2}{\omega^2} \operatorname{Im}\left[ \frac{1}{Z_A(\omega)} \right], \tag{32}$$

for the gluon spectral function and

$$\rho_c(\omega) = \frac{\pi}{Z_c} \delta(\omega^2) - \frac{2}{\omega^2} \operatorname{Im}\left[ \frac{1}{Z_c(\omega)} \right], \tag{33}$$

for the ghost spectral function.

The combination of (31), (32), (33) allows us to compute both gluon and ghost spectral functions $\rho_A$ and $\rho_c$ as well as the respective propagators for complex frequencies, and in particular for spacelike (Euclidean) and timelike frequencies.

### 3.5 Iterative procedure

The spectral DSEs for ghost and gluon propagator (31) are solved using an iteration procedure, discussed in detail in [1], and briefly reviewed below:

Assuming spectral representations for ghost and gluon propagator, the gluon spectral function $\rho_A^{(i)}$, obtained after the $i$-th iteration step with input $\rho_c^{(i)}$, is inserted together with $\rho_c^{(i)}$ into the spectral integral form of $\Sigma_{\bar{c}c}(p)$, on the right-hand side of (31b). Then, by means of (33), we arrive at the $(i+1)$-th ghost spectral function, $\rho_c^{(i+1)}$. In turn, $\rho_c^{(i+1)}$ is then inserted together with $\rho_A^{(i)}$ into the spectral integral form of $\Sigma_{AA}(p)$, on the right-hand side of (31a). With (32), we then obtain $\rho_A^{(i+1)}$. This iteration is repeated until simultaneous convergence for both spectral functions has been reached. The iteration commences with initial choices for $\rho_A$ and $\rho_c$. Along with convergence properties, these choices are discussed in Appendix H.

Attempts to solve the system for $\rho_A$ and $\rho_c$ via a Newton's optimization scheme in a purely spectral manner should worse convergence properties than the iterative approach. For this reason, the optimization approach was not pursued further.

# 4 Complex structure of Yang-Mills theory with complex conjugate poles

In this section, we analytically show that a gluon propagator with a simple pair of complex conjugate poles cannot be part of a consistent solution of the coupled DSE system for Yang-Mills propagators in the Landau gauge with bare vertices. This analysis is presented in detail in Appendix D.1 and Appendix D.2, and makes use of the spectral DSE set up in Section 3. Note that, while building on spectral representations, the spectral DSE is also suitable for studying propagators violating the ordinary KL representation, but obeying more general spectral/integral representations including complex poles.

Before presenting the conclusions of this analysis, we provide a brief overview of results on spectral representations in YM theory and discuss the manifestation of single pairs of complex conjugate poles in Section 4.1. This is followed by a discussion of the generic impact of singularities in coupled sets of functional equations as well as the requirements for conclusive studies in Section 4.2.

## 4.1 Complex structure of Yang Mills propagators

The complex structure of the Yang-Mills propagator, and specifically the gluon propagator, is the subject of an ongoing debate. Axiomatic formulations of local QFTs forbid the existence of any further non-analytic structures beyond the real frequency axis for propagators of asymptotic states. It has been argued that this also applies to gauge theories, and in particular the case of the gluon propagator [39, 41, 80]. Scenarios such as complex conjugate poles are nevertheless used in reconstructions of the timelike structure of the gluon propagator, see e.g. [6, 10, 12, 37, 38, 40, 42–45]. However, precision reconstruction of Yang-Mills propagators in a purely spectral manner and without complex conjugate poles has successfully been performed in [3, 4, 8, 81]. Gauge dependence of spectral functions and analytic properties of propagators has been studied, e.g., in [82, 83].

In Appendix D.1 and Appendix D.2 we investigate the consequences of a single pair of complex conjugate poles in the gluon propagator on the complex structure of Yang-Mills theory *fully analytically*. While being not fully general, this scenario represents the simplest and so far only considered case of violation of the spectral representation, both in reconstructions and analytic considerations.

The spectral formulation employed in Section 3 enables us to study the general complex structure of ghost and gluon DSE, as it covers a large class of functions for the propagators and is by no means restricted to propagators satisfying the KL representation (7). In particular, a gluon propagator with a pair of complex conjugate poles is realised by collapsing the (gluonic) spectral integrals at complex spectral values corresponding to complex conjugate pole positions, multiplied by the respective residues. Within the iterative approach to solving DSEs described in Section 3.5, we are able to track the propagation of these non-analyticities through the iterations of ghost and gluon DSE. This is done in an expansion about the fully analytic spectral parts of all the diagrams. In other words, we only consider the contributions arising from adding the holomorphicity violating complex conjugate pole part of the gluon propagator.

Explicitly, for both, ghost and gluon, propagators we will employ the parametrisation

$$G = G^{\text{KL}} + G^{\chi} \,. \tag{34}$$

The non-spectral part $G^{\chi}$ encodes the respective violation of the KL representation, either directly given by the complex conjugate poles as for the gluon, or for the ghost induced by the complex conjugate poles. The spectral contribution $G^{\text{KL}}$ is given by the KL representation (7)

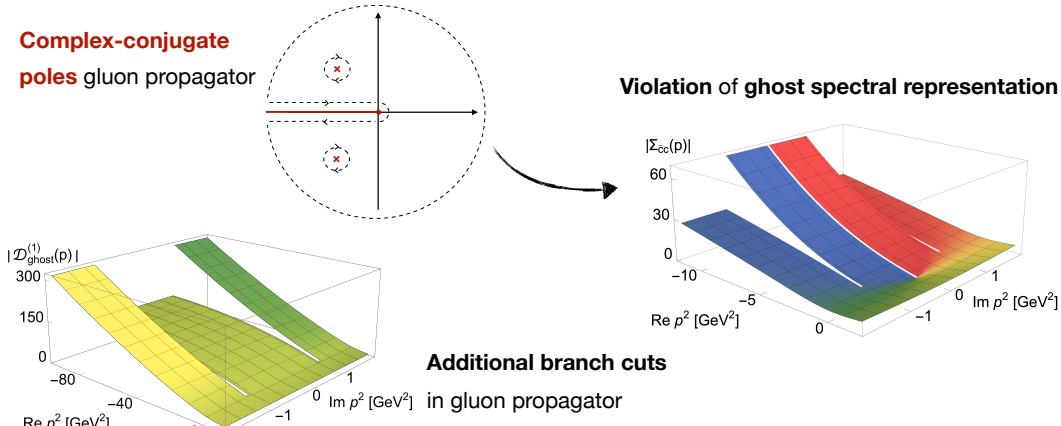

Figure 4: Propagation of non-analyticities in the coupled Yang-Mills system with bare vertices. The displayed calculation is *fully analytic*. Complex poles in the gluon propagator cause additional branch cuts off the real axis in the ghost propagator, as shown in the plot on the right (see Figure 11 for full size). Hence, the Källén-Lehmann representation of the ghost is violated. These additional branch cuts generate corresponding additional branch cuts also in the gluon propagator via the ghost loop, demonstrated in the bottom left figure (see Figure 13 for full size). This violates the initial assumption of just a single pair of complex poles in the gluon propagator. In consequence, a single pair of complex poles cannot feature alone in consistent solutions in our truncation. The explicit analytic computation is presented in Appendix D.

of the respective propagator. With the spectral-non-spectral split (34), the contributions to the single diagrams can be ordered in powers of non-spectral contributions $G^\chi$ entering. We only consider one-loop diagrams with two propagators in the spectral DSE setup of Section 3. Hence, the contributions coming from the additional non-analyticities that we will consider here are given by $G^{KL}G^\chi$, $(G^\chi)^2$. The ordinary spectral part is constituted by $(G^{KL})^2$.

## 4.2  Propagation of non-analyticities

The systems of DSEs are integral equations, typically solved within an iterative procedure. In such an iteration, non-analyticities off the real frequency axis propagate through the system by the iteration. Here, we use this mechanism to study if complex poles allow for an analytically consistent solution to Yang-Mills theory. Our main results can be summarised as follows:

*In Yang-Mills theory with bare vertices, a pair of complex poles in the gluon propagator*

1. *violates the Källén-Lehmann representation of the ghost and*

2. *cannot be part of an analytically consistent solution of Yang-Mills theory without additional branch cuts in the complex plane.*

These results are obtained by the following analysis: We assume a gluon propagator with only a single pair of complex poles. Via the ghost self-energy diagram, these poles induce additional branch cuts off the real frequency axis in the ghost propagator. Hence, the spectral representation of the ghost propagator is violated. The additional cuts in the ghost propagator can be represented via a modified spectral representation. We use this representation to study the back-propagation of these additional cuts into the gluon propagator via the ghost loop of the gluon DSE. There, we observe that the cuts likewise induce branch cuts off the real

frequency axis in the gluon propagator. This is at odds with the initial assumption of a single pair of complex conjugate poles. A consistent solution in the above scenario, involving a single pair of complex conjugate poles as well as bare vertices, is hence ruled out: An analytically consistent solution at least needs to be accompanied by the respective pair of branch cuts. The explicit calculation is carried out in Appendix D. We visualise the propagation of non-analyticities in the system in Figure 4. Note also that the performed analysis is independent of possibly different infrared scenarios such as scaling, decoupling or massive solutions.

If the non-trivial vertices do not annihilate the additional complex singular structures, this mechanism readily carries over to the full Yang-Mills system. The former annihilation either requires a respective ghost-gluon vertex that counteracts the loss of the spectral representation of the ghost, or combinations of diagrams and vertices in the gluon gap equation prohibit the back-propagation of the additional branch cuts of the ghost.

While a full analysis goes far beyond the scope of the present work, we briefly evaluate the above-mentioned simplest possibility: a non-trivial complex structure in the classical dressing of the ghost-gluon vertex that counteract the effects of complex conjugate poles in the gluon propagator in the ghost DSE. This is seemingly reminiscent of the cancellation of complex poles in the electron propagator in QED: There one can solve the electron gap equation under the assumption, that the photon enjoys a spectral representation. Then, the solution of the electron gap equation with bare electron-photon vertices leads to complex conjugate poles for the electron. These artefacts disappear if dressed vertices are used, that satisfy the Ward-Takahashi identity. The latter vertex dressings are proportional to differences of the wave functions of the electrons, balancing the (inverse) wave function in the propagator.

This mechanism in the electron gap equation in QED does *not* apply to the ghost DSE in QCD. First, the ghost shows additional branch cuts, not complex poles as the electron propagator in the scenario discussed above. Second, these branch cuts are due to complex poles in the gluon propagator, which was shown in [2]: There, it was shown that using a spectral gluon propagator and bare vertices, complex poles are absent in the ghost, and the spectral representation is intact. Furthermore, no sign for a loss of the ghost spectral representation has been hinted at in all investigations so far. A cancellation of the complex singularities of the gluon in the ghost gap equation hence needs to involve the ghost-gluon vertex's scattering kernel that is usually left out in the STI construction. We consider such a delicate balance scenario as unlikely, and it has no counterpart in similar or seemingly similar systems in the literature.

Note that this assessment is merely an interpretation of our structural results. We emphasise that a conclusive analysis of the complex structure of the Yang-Mills system requires a fully non-perturbative study, as the dynamical emergence of the gluon mass gap is non-perturbative. It is difficult to envisage such a fully analytical study in the near future, and a numerical study almost by definition has to rely on approximations and hence lack a fully conclusive nature. This is already evident from the present study, as we only can exclude complex conjugate poles in the present approximation.

The above arguments emphasise the difficulty of studies in Yang-Mills theories, so one may first study variants thereof: In the past decade many studies have also exploited massive extensions of Yang-Mills, as for example formulated in terms of the Curci-Ferrari (CF) model with mass terms for ghosts and gluons [84–86]. This approach only constitutes a model for Yang-Mills theory due to the presence of an additional relevant parameter, the gluon mass, and the almost certain lack of unitarity at least for large values of the explicit gluon mass parameter. Still, it offers an analytic way for studying part of the full problem, which already has proven useful. For an in-depth differentiation between the CF model and YM theory, see Appendix A.

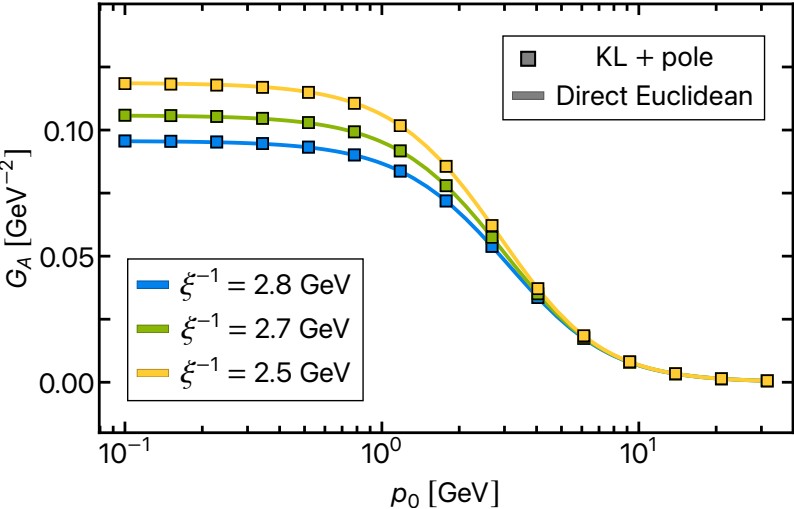

Figure 5: Gluon propagator for different values of $m_A^2 = -2.98, -1.24$ and $-0.31\,\text{GeV}^2$. Solid lines represent the propagators computed directly via the spectral Euclidean DSE (29). The squared points are obtained by a sum of the spectral contribution $G_A^{\text{KL}}$ and the fit $G_A^{\text{approx}}$ of the spectral difference $\Delta G_A$ defined in (35). $G_A^{\text{KL}}$ is computed from the real-time DSE via the spectral representation (7), while the fit $G_A^{\text{approx}}$ is constituted by a pole on the real frequency axis, see (37). All propagators are of decoupling type, as they become constant in the IR. Their asymptotic value increases with decreasing $m_A^2$. The propagators have been rescaled to lie on top of each other in the perturbative region, cf. Appendix F.

In a massive extension of Yang-Mills theory, complex conjugate poles may occur in the gluon propagator at one-loop. This implies that their impact on the ghost propagator may be visible at two-loop in the ghost gap equation. Accordingly, the back-propagation of the ghost propagator's non-analyticities into the gluon DSE at least requires a perturbative three-loop computation. While certainly being challenging, this may be within the technical range of perturbative computations in the CF model, and is very desirable. The back-propagation of the additional cuts poses a major obstruction that only can be circumvented by intricate relations between the complex structures of propagators and that of the vertices, in particular the ghost-gluon vertex. Signs for the latter gathered in perturbation theory at least require a three-loop analysis of the ghost DSE as argued in Section 4.2. Such an analysis, while highly desirable, has not been undertaken yet in the literature.

To wrap up, direct or reconstructed solutions with complex conjugate poles and additional cuts should undergo a self-consistency analysis as presented in this section before being considered further. On the constructive side, the present self-consistency considerations of the complex structure can be used to devise self-consistent spectral or generalised spectral representations for correlation functions, either generic ones or restricted to a given approximation at hand.

# 5 Numerical results

In this section, we present numerical solutions of the coupled system of spectral ghost and gluon propagator DSEs of Yang-Mills theory set up in Section 3. A common solution strategy in functional approaches is to gradually tune the gluon mass parameter $m_A^2$ towards the confining region of the theory, indicated by an increasing gluon mass gap with decreasing $m_A^2$; see,

e.g., [46,78] and [47] for applications in the fRG and DSE, respectively. In the gluon DSE, the mass parameter enters via the mass counterterm in case of quadratic divergences, as in (27) for the case of spectral BPHZ renormalisation. Strictly speaking, by the presence of an additional parameter the described approach only constitutes a model of Yang-Mills theory, additionally breaking gauge invariance. In [47] it was shown that such a parameter can be allowed for in the case where it can be understood as a sign of a dynamical generation of a gluon mass gap, however. This scenario entails the presence of irregular vertices, however, as is the case in the Schwinger mechanism [47,87–94] or the scaling solution of YM [46,55,76,78,95]. The scaling solution has the appeal of singling out a unique value of for the $m_A^2$, eliminating the additional parameter again. While the gluon mass parameter a priori breaks gauge invariance, the amount of gauge symmetry breaking for particular solutions corresponding to different choices of $m_A^2$ is encoded in the STIs. Monitoring the strength of the violation of gauge symmetry as well as its potential restoration in the confining regime hence requires to also solve the longitudinal sector of the theory, as done in [78], but is beyond the scope of the present work.

In the present work, we follow above described strategy of tuning the mass parameter towards the confining region in order to study the timelike structure of ghost and gluon propagators. Note that this procedure has to be carefully distinguished from the usual procedure applied in the Curci-Ferrari model, where commonly, the ghost and gluon gap equations are evaluated at a given loop order with fixed ghost and (massive) gluon propagator input. Furthermore, often times a phenomenological interpretation is assigned to the gluon mass parameter in the CF model, in contradistinction to Yang-Mills theory. In functional approaches, the system is solved self-consistently, with full propagators inside the loops. The procedure described aiming at eliminating the introduced gluon mass parameter and restoring BRST invariance. In conclusion, while both approaches introduce a gluon mass parameter in the first place, the main difference lies in its interpretation and how it is dealt with in regard of BRST invariance, as outlined in above paragraph.

The numerical solutions within our spectral DSE setup are obtained by iteration, starting with an initial choice for the gluon and ghost spectral functions $\rho_A$ and $\rho_c$. Then, the coupled system of ghost and gluon gap equations is solved self-consistently for a family of input gluon mass parameters $m_A^2$. The value of the renormalisation scale is set to $\mu_{\mathrm{RG}} = 5$ internal units (i.u.), which is converted to physical units as described in Appendix F. This yields a slightly different renormalisation scale $\mu_{\mathrm{RG}}$ for each input parameter $m_A^2$, which is always around $\mu_{\mathrm{RG}} \approx 10$ GeV. The renormalisation conditions specified in (28a) are employed.

## 5.1 Spectral violation

A simple and analytically consistent scenario for ghost and gluon propagator involves solely simple branch cuts on the real axis for both, and a massless pole for the ghost. This leaves their KL representation intact, see Section 3, and allows to solve the system iteratively in a fully spectral manner, cf. Section 3.5. Our attempts to find such a fully spectral solution were plagued by violations of the gluon spectral representation, however. In particular, we could not find an initial guess for the gluon spectral function which did not violate the KL representation in the gluon DSE (21). The violation of the spectral representation can be assessed by subtracting the spectral propagator from the directly computed one, i.e.

$$\Delta G_A(p) = G_A(p) - G_A^{(\mathrm{KL})}(p).$$ (35)

Here, $G_A$ is the propagator obtained directly from the real- and imaginary-time DSEs (31) and (29), while $G_A^{\mathrm{KL}}$ is calculated from the gluon spectral function (32) obtained from the spectral DSE (31).

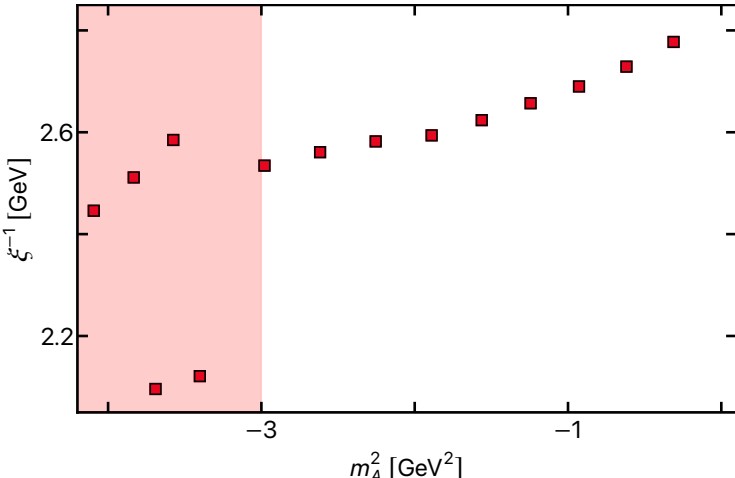

Figure 6: Screening lengths of the gluon propagators as defined in (38) for the whole family of solution as a function of the input gapping parameter $m_A^2$. The drastic, non-monotonic change of the screening length to the left of $m_A^2 \approx 3\,\text{GeV}^2$ hints at numeric instabilities in the solutions. These are most like induced by worsening of the spectral difference approximation (37). Therefore, all solutions in the red shaded region will neither be presented nor discussed.

If $\Delta G_A$ is non-zero, the spectral representation is violated, and we found $\Delta G_A \neq 0$ for all our initial guesses. In consequence, the corresponding gluon propagator must exhibit further complex structures such as (one or more pairs of) complex conjugate poles or further branch cuts in the complex plane, which violate the spectral representation. In fact, in all cases the spectral difference $\Delta G_A$ is fit quite well by a single pair of complex conjugate poles, suggesting that the violation is mainly due to a single pair of these poles. It thus seems natural to just include these additional complex poles into our approach. However, this comes along with several problems: First, in order to directly resolve these non-analytic structures and precisely determine their position, one would have to resolve the full complex momentum plane. While in the fully spectral approach, only the Euclidean and Minkowski axis have to be resolved, evaluating the DSEs in the full complex plane would drastically increase the numerical effort. Most importantly though, the analytic solutions of the momentum loop integrals presented in Appendix B are a priori not valid for arbitrary complex momenta $p$ and complex masses $\lambda$. This issue is further discussed in Appendix B.4.

Last but not least, from the findings of Section 4 it becomes evident that a self-consistent solution of the coupled YM system with a pair of complex conjugate poles in the gluon propagator and a spectral ghost propagator is not possible with just bare vertices. The complex conjugate pole part of the gluon propagator directly induces two additional branch cuts in the complex plane for the ghost propagator, see Figure 11. While these can be captured via a modified spectral representation as in (D.6) and shown in Figure 12, the additional cuts in the ghost propagator in turn induce (at least) two further branch cuts in the gluon propagator via the ghost loop, see Figure 13. In consequence, a pair of complex conjugate poles for the gluon propagator evidently leads to a cascade of additional non-analytic structures for both ghost and gluon propagator. This renders a consistent solution of the full theory including such a pair of poles highly improbable.

Note that complex conjugate poles appear generically at the one-loop level of massive extensions of Yang-Mills. This already suggests that our solutions are in the Higgs-type branch of the theory, where we do not necessarily expect a spectral representation of the gluon propagator. This is supported by the form of the gluon propagator in Figure 5, as well as the relation

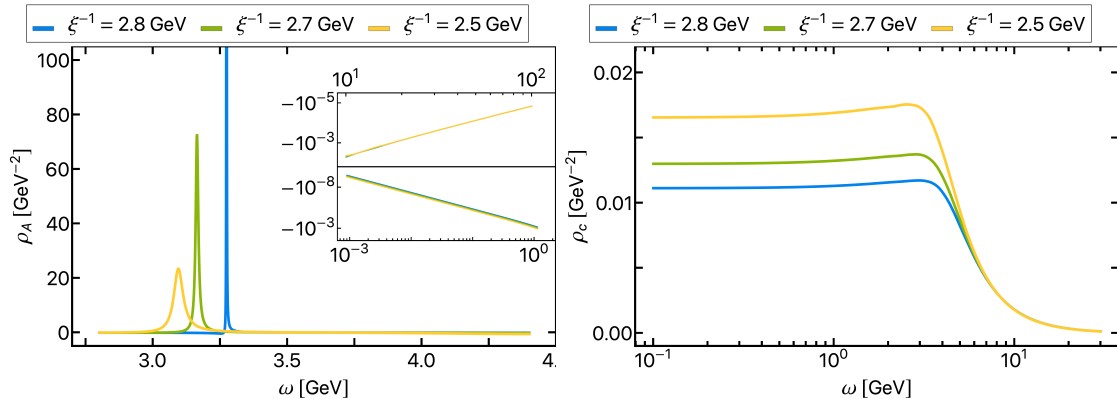

Figure 7: Gluon (left) and ghost (right) spectral functions for different inverse screening lengths, corresponding to the values of the input gapping parameter $m_A^2 = -2.98, -1.24$ and $-0.31\,\mathrm{GeV}^2$. For decreasing inverse screening length, the peak amplitudes of the gluon spectral function decreases significantly and a second negative peak at larger frequencies becomes more pronounced. The inset shows that both IR and UV tail of all gluon spectral functions approach the axis from below. As discussed in Section 2, this property can be derived analytically by demanding a Källén-Lehmann representation for the gluon propagator. Although our gluon propagator minimally violates the spectral representation (comp. Figure 8), we still find the negativity of both asymptotic tails to hold. The ghost spectral function $\rho_c$ shown in the right panel varies only in magnitude under variation of $m_A^2$. All ghost spectral functions coincide w.r.t. to shape. In particular, they show a constant behaviour for $\omega \to 0$, which is a manifestation of the purely logarithmic branch cut of the ghost propagator. For larger frequencies, the ghost spectral functions approach zero.

between the screening mass $\xi^{-1}$ (not to be confused with the gauge fixing parameter) and the input mass parameter $m_A^2$ in Figure 6. Ultimately, we are interested in the confining branch of the theory. In order to reach this branch, the system needs to be tuned in this direction via variation of the input parameter $m_A^2$, for a detailed discussion see c.f. [46, 47].

If the discrepancy $\Delta G_A$ in (35) is non-zero, the spectral part of the gluon propagator with the spectral function $\rho_A$ as defined in (32) does not account for the full gluon propagator $G_A$ any more, as discussed above. In order to still feed back an on both axes well approximated gluon propagator, we also need to feed back the spectral difference $\Delta G_A$. We approach this via a fit. The fit Ansatz for $\Delta G_A$ is required to avoid the above described cascade of non-analyticities induced by complex conjugate poles, while approximating the numerically given spectral difference (35) as good as possible. Firstly, we note that $\Delta G_A$ is a purely real quantity, as $\mathrm{Im}\,G_A^{\mathrm{KL}} = \mathrm{Im}\,G_A$ due to

$$
\begin{aligned}
\mathrm{Im}\,G_A^{\mathrm{KL}}(-\mathrm{i}(\omega + \mathrm{i}0^+)) &= \mathrm{Im}\!\left[ \int_\lambda \frac{\rho(\lambda)}{-(\omega + \mathrm{i}0^+)^2 + \lambda^2} \right] \\
&= \frac{\rho^A(\lambda)}{2} = \mathrm{Im}\,G_A(-\mathrm{i}(\omega + \mathrm{i}0^+)),
\end{aligned}
\tag{36}
$$

where in the last line we used the Sokhotski-Plemelj theorem. Note that $\Delta G_A$ can generally be only computed at frequencies $p$ where the gluon DSE (29) is evaluated. In our case, these are either purely real or imaginary frequencies.

As discussed in Section 4, the spectral DSEs set up in Section 3 are able to account for propagators with real or complex poles or Källén-Lehmann-like integral representation, such as the

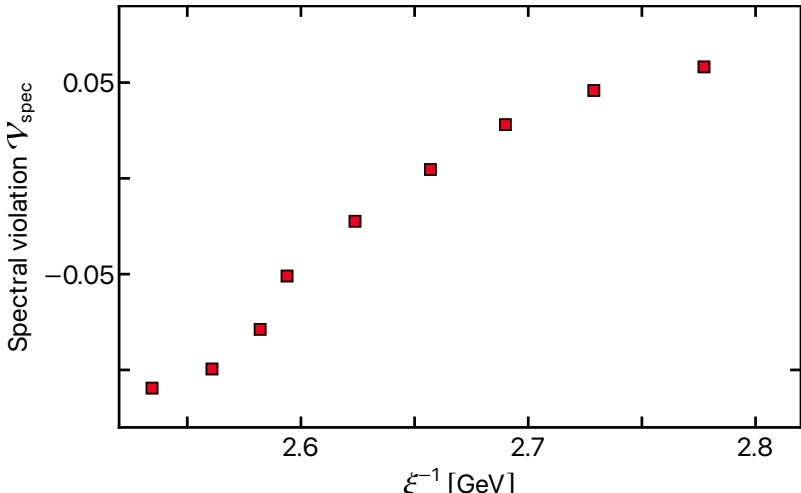

Figure 8: Spectral violation of the gluon propagators for the whole family of solutions as a function of the inverse screening length. In the considered interval, the spectral violation exhibits a root at $2.65 < \xi^{-1} < -2.7$ GeV. Hence, the total weight of all non-analyticities flips sign.

modified spectral representation for the ghost (D.6). Incorporating $\Delta G_A$ into our calculation can be achieved by modelling $\Delta G_A$ by a pole on the real frequency axis,

$$\Delta G_A(p) \approx G_A^{\mathrm{approx}}(p) = \frac{Z_\chi}{p^2 + \chi^2}, \tag{37}$$

with real $\chi > 0$. We emphasise that the parametrisation (37) of the spectral difference by a pole on the real frequency axis solely constitutes a convenient approximation of all non-holomorphicities of the gluon propagator beyond its branch cut on the real frequency axis. In particular, due to the existence of a branch cut on the real frequency axis, if such a pole existed it would directly show up as a singularity in the spectral function. This is not the case, however.

In explicit, adding $G_A^{\mathrm{approx}}$ to the Källén-Lehmann part $G_A^{(\mathrm{KL})}$, in (29) resp. (26) we simply substitute $\rho_A(\lambda) = \rho_A^{(\mathrm{KL})}(\lambda) + Z\,\delta(\lambda^2 - \chi^2)/\pi$, where $\rho_A^{(\mathrm{KL})}$ is still given by (32).

## 5.2 Numerical solutions

Accounting for spectral violations with the procedure described in Section 5.1, the coupled DSE system of Yang-Mills theory is solved with a strong coupling constant $\alpha_s = 0.2$ for a family of input gapping parameters $m_A^2 \in [-3.69, -0.31]$. For values of $m_A^2$ beyond this region, we were not able to converge to a solution. The solutions corresponding to the different numerical inputs $m_A^2$ are labelled by the respective (inverse) screening lengths of the gluon propagators instead, which are related to the gluon mass gap.

The temporal screening length $\xi$ (not to be confused with the gauge fixing parameter) is defined through the Fourier transform of the propagator in momentum space $G(p)$, which is

$$\lim_{|x_0 - y_0| \to \infty} \int_{-\infty}^{\infty} \frac{\mathrm{d}p_0}{2\pi}\, e^{ip_0(x_0 - y_0)}\, G(p) \sim e^{-|x_0 - y_0|/\xi}. \tag{38}$$

According to (38), the screening length $\xi$ governs how fast the propagator decays at large temporal distances. Here, it is readily evaluated by computing the Fourier transforms of the Euclidean propagators and determining $\xi$ via an exponential fit of the large distance behaviour.

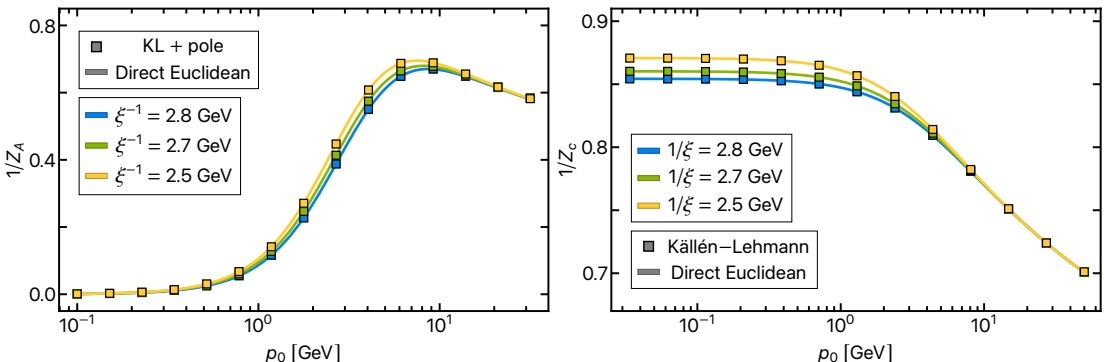

Figure 9: Gluon (left) and ghost (right) dressing functions for different values of the input gapping parameter $m_A^2$. Solid lines represent the dressing functions computed directly via the spectral Euclidean DSE (29). In case of the gluon (left), the squared points are obtained by a sum of the dressing corresponding to the spectral contribution $G_A^{KL}$ and the fit $G_A^{approx}$ of the spectral difference $\Delta G_A$ defined in (35). $G_A^{KL}$ is computed from the real-time DSE via the spectral representation (7), while the fit $G_A^{approx}$ is constituted by a pole on the real frequency axis, see (37). For the ghost (right), the squared points are given solely by the spectral representation of the dressing. For decreasing $m_A^2$, the peak position of the gluon dressing function moves towards smaller frequencies. The ghost dressing functions shown in the right panel are of decoupling-type and become constant in the IR. barely vary and change of $m_A^2$. All dressing functions have been rescaled to lie on top of each other in the perturbative region, cf. Appendix F.

The gluon propagators' inverse screening length as a function of the $m_A^2$ is shown in Figure 6, and decreases monotonically with decreasing $m_A^2$ for all solutions considered here.

Since our self-consistent Yang-Mills system does not have inherent scales, we set the scale by rescaling all solutions to coincide with the fRG Landau gauge Yang-Mills data of [46] in the deep perturbative region; details can be found in Appendix F.

The resulting gluon spectral functions $\rho_A$ are shown in Figure 7 for $m_A^2 = -2.98, -1.24, -0.31 \, \text{GeV}^2$. For larger $m_A^2$, the gluon spectral function develops a strong and very sharp positive peak. At the lower end of the family of solutions w.r.t $m_A^2$, the gluon spectral function develops a slight negative peak at around 4 GeV, while generally the peak amplitudes decreases a lot. The inset in the left panel of Figure 7 shows that both IR and UV tail of all gluon spectral functions approach the axis from below. As discussed in Section 2, this property can be derived analytically by demanding a Källén-Lehmann representation for the gluon propagator. Although our gluon propagator minimally violates the spectral representation (comp. Figure 8), we still find the negativity of both asymptotic tails to hold.

However, all gluon propagators presented in Figure 5 feature a spectral violation, see Section 5.1. This means that the spectral functions displayed in the left panel of Figure 7 do not make up for the whole propagator. In order to quantify the size of the gluon propagator's fraction constituted by the spectral part $G_A^{KL}$, we define the spectral violation

$$\mathcal{V}_{\text{spec}} = \frac{1}{\|G_A\|_{\mathcal{L}_1}} \int_0^\infty \mathrm{d}p \left( G_A^{KL}(p) - G_A(p) \right). \tag{39}$$

Note that only approximately $G_A \approx G_A^{KL} + G_A^{approx}$ due to (37), which is why we leave the difference $G_A^{KL} - G_A$ in (39) explicit.

The spectral violation (39) as a function of the screening length is visualised in Figure 8 for all solutions. We find that the (magnitude) of the spectral violation is increasing towards the boundary of the $m_A^2$-interval for which we are able to solve the system. The fact that convergence worsens for large spectral violation can be attributed to the fact the spectral difference $\Delta G_A$ is only approximately taking into account via a pole on the real frequency axis (37). The larger the absolute value of the spectral violation $\mathcal{V}_{\text{spec}}$ gets, the larger the approximation error gets. A more in-depth discussion of the quality of the approximation, in particular on the real frequency axis, is deferred to Appendix G.

Inspecting the shape of the gluon propagators presented in Figure 5, we find that the value of the gluon propagator in the origin increases with decreasing $m_A^2$, which signals the Higgs-type branch of our solutions. In short, none of our solutions is in the confining region, for more details see [46, 47, 78, 96]. In consequence, a statement about the complex structure of Yang-Mills in the confining phase within the chosen approximation cannot be made.

The ghost spectral functions of the presented solutions are shown in the right panel of Figure 7. Evidently, the change of $\rho_c$ under a variation of $m_A^2$ is much smaller. All ghost spectral functions coincide with respect to shape. In particular, they show a constant behaviour for $\omega \to 0$, which is a manifestation of the purely logarithmic branch cut of the ghost propagator. For larger frequencies, the ghost spectral functions approach zero. In summary, these results agree qualitatively very well with our previous studies of the ghost spectral function, which have been carried out via the stand-alone spectral ghost DSE in [2] and via reconstruction of QCD lattice data with Gaussian process regression in [8].

The corresponding gluon and ghost dressing functions are shown in Figure 9. For decreasing $m_A^2$, the peak position of the gluon dressing function moves towards smaller frequencies. In order to assess how well the approximation of the spectral difference (35) as a single particle pole (37) works, we compare the dressing computed directly via the spectral Euclidean DSE (29) against the one given by the sum of the spectral part and the fit of the spectral difference part, $G_A^{\text{KL}} + \Delta G_A$. It can be seen that the dressings match very well, supporting the single pole approximation for the shown Euclidean solutions. In case of the propagators, see Figure 5, the comparison is more sensitive to the IR. Also there, single pole approximation works reasonably (on the Euclidean branch). For an in-depth discussion of the approximation on the Minkowski axis, see Appendix G. The ghost dressing functions accordingly are also all of decoupling-type as they become constant in the IR. For decreasing $m_A^2$, the IR value of the ghost dressing function increases. Here, the spectral representation is intact.

## 6 Conclusion

In this work, we investigated the complex structure of Yang-Mills theory with the help of the spectral Dyson-Schwinger equation. Our approach is based on [1] and utilises the spectral renormalisation scheme devised there. In particular, the spectral DSE allows for analytic solution of the momentum loop integrals of all involved diagrams. As a result, we gain direct analytic access to the complex structure of ghost and gluon propagator.

In Section 4, we studied the analytic structure of Yang-Mills theory with bare vertices and a gluon propagator with complex conjugate poles. Our findings could hint at the fact that a self-consistent solution of Yang-Mills is not possible with a gluon propagator featuring one or more pairs of complex conjugate poles. Specifically, we were able to show analytically in the case of bare vertices, that a self-consistent solution with complex conjugate poles and no further branch cuts does not exist. Complex conjugate poles in the gluon propagator directly violate the spectral representation of the ghost propagator by two additional branch cuts off the real axis. This, in turn, introduces additional branch cuts off the real axis in the gluon

propagator via the ghost loop. These further cuts contradict the initial assumption of single pair of complex conjugate poles. The study hence shows that by seeding complex singularities in the gluon propagator, a cascade of non-analyticities is induced, which propagate through the system by iteration. Eventually, this observation could disfavour Yang-Mills solutions with complex conjugate poles and no further branch cuts in the complex plane. We emphasise that this analytic result is independent of the different solution 'branches' of Yang-Mills such as scaling, decoupling or massive.

A central aspect of our analytic study of the complex structure of Yang-Mills theory in Section 4 is, that the existence of complex conjugate poles in the gluon propagator leads to a violation of the spectral representation for the ghost, at least for the case of bare vertices. For this not to carry over to full YM theory, an intricate cancellation of the complex poles in the gluon propagator by the full ghost-gluon vertex is required. In our opinion, this is unlikely to occur in Yang-Mills theory or QCD. In particular, a respective analysis requires at least three-loop consistency. We remark that no sign of a violation of the spectral representation has been found for the ghost propagator in various works [6,7,12,97]. Therefore, our results emphasise the need for analysing consistency of analytic structure in particular in results with complex conjugate poles for the gluon propagator in QCD like regions.

In Section 5, we iteratively solve the coupled system of spectral DSEs for the YM propagators at real and imaginary frequencies. We find decoupling-type solutions for which the Källén-Lehmann representation of the gluon propagator is partially violated, depending on the choice of input gapping parameter. The gluon spectral functions obey the known analytic constraints on the asymptotic behaviour. Solving the system for more QCD-like regions is hindered by increasing violation of the spectral representation, which is accounted for approximatively.

The analytic structure of Yang-Mills theory therefore remains unclear: In Section 4 we present an analysis implying that for a consistent solution with complex poles in full YM theory, a delicate cancellation in the analytic structure of propagators and vertices would need to happen. As we were able to show, with bare vertices, such a solution without further cuts is even ruled out. On the other hand, in our numerical study in Section 5 we were not able to solve the system with allowing for violation of the KL representation of the gluon. We observed the generic appearance of complex poles for a vast range of initial conditions. Hence, a conclusive statement about the complex structure of Yang-Mills in the confining region based on the present results is not possible. However, the current work lays the foundation for such an analysis, and we hope to report on the respective results in the near future.

## Acknowledgments

We thank G. Eichmann, J. Papavassiliou and U. Reinosa for discussions. This work is done within the fQCD-collaboration [98], and we thank the members for discussion and collaborations on related projects.

**Funding information**   This work is funded by the Deutsche Forschungsgemeinschaft (DFG, German Research Foundation) under Germany's Excellence Strategy EXC 2181/1 - 390900948 (the Heidelberg STRUCTURES Excellence Cluster) and under the Collaborative Research Centre SFB 1225 (ISOQUANT) and the BMBF grant 05P18VHFCA. JH acknowledges support by the GSI FAIR project and Studienstiftung des deutschen Volkes. NW is also supported by the Hessian collaborative research cluster ELEMENTS and by the DFG Collaborative Research Centre "CRC-TR 211 (Strong-interaction matter under extreme conditions)".

# A    Difference between Yang-Mills theory and the Curci-Ferrari model

We would like to expand on the difference between Yang-Mills theory and the Curci-Ferrari model. For a recent review of the CF model see [99]. Here, we focus on issues related to the (non-perturbative) IR completion of the Landau gauge, ignoring other differences such as the debate whether or not the CF model is unitary for some range of gluon masses. The CF model features a free gluon mass parameter in the classical action, cf. (1). YM theory, on the other hand, has no free parameter. This leads to a modified BRST symmetry, sometimes called mBRS symmetry, in the CF model, which is not nilpotent anymore [84].

Our current way of dealing with YM theory in Functional Approaches, and specifically in both the fRG and DSE, requires the violation the Slavnov-Taylor identities by either the regularization and/or the truncation. The resolution of this conundrum is well understood. The breaking of the related symmetry introduces a new and relevant direction at the UV fixed point. This novel direction has the largest overlap with the mass parameter of the gluon, i.e. we cannot ignore this parameter. However, it is uniquely fixed by satisfying the STIs in parallel to the DSE/fRG equations. For computational reasons, we are limited in the present setup, extending to Minkowski space-time, to the transverse part of Landau gauge and we cannot use the STI constraints directly to fix the parameter. It has been argued, that the scaling solution is one possible solution to the IR closure of Landau gauge. Additionally, the gluon mass parameter can also be used to tune the system towards scaling, and hence resolve the STIs in addition. For detailed discussions on this issue we refer the interested reader to [46,47], where scans of the mass parameter in fRG and DSE approaches respectively have been performed, and to [78] for a combined discussion of tuning of the mass parameter and the STIs.

In practice, this leads to a strong similarity between YM theory in Landau gauge and the CF model. In sufficiently crude truncations, such as in this work, the systems could be interpreted in the view of both. Here, we are trying to get insight into the spectral functions of Yang-Mills theory, and hence interpreted all our results from the appropriate perspective. This includes the handling of the mass parameter, which we tuned as close to scaling as computationally feasible.

# B    Loop momentum integration

In this appendix we detail the analytic solution of the loop momentum integrals of the self energy diagrams (26) of the spectral DSEs (21) at the example of the ghost self energy diagram $\Sigma_{\bar{c}c}$. Starting at (25), we express the ghost-gluon diagram as

$$\Sigma_{\bar{c}c}(p) = g^2 \delta^{ab} C_A \int_{\lambda_1,\lambda_2} \rho_A(\lambda_2) \rho_c(\lambda_2) I(p,\lambda_1,\lambda_2), \tag{B.1}$$

with the now dimensionally regularised momentum integral

$$I(p,\lambda_1,\lambda_2) = \int_q \left( p^2 - \frac{(p \cdot q)^2)}{q^2} \right) \frac{1}{q^2 + \lambda_1^2} \frac{1}{(p+q)^2 + \lambda_2^2}. \tag{B.2}$$

The measure is now $\int_q = \int d^d q/(2\pi)^d$.

## B.1 Momentum integration

Next, we employ partial fraction decomposition

$$\frac{1}{q^2}\frac{1}{q^2+\lambda^2} = \frac{1}{\lambda^2}\Big(\frac{1}{q^2} - \frac{1}{q^2+\lambda^2}\Big), \tag{B.3}$$

and introduce Feynman parameters, i.e. utilise

$$\frac{1}{AB} = \int_0^1 dx \frac{1}{xA+(1-x)B}. \tag{B.4}$$

Upon a shift in the integration variable $q \to q - xp$ and after some manipulation, we arrive at

$$I(p,\lambda_1,\lambda_2) = \int_{q,x} \sum_{i=0}^2 (q^2)^i \bigg[\frac{A_i}{(q^2+\tilde{\Delta}_1)^2} + \frac{B_i}{(q^2+\tilde{\Delta}_2)^2}\bigg], \tag{B.5}$$

with

$$\tilde{\Delta}_1 = (1-x)\lambda_1^2 + x\lambda_2^2 + x(1-x)p^2,$$

$$\tilde{\Delta}_2 = \tilde{\Delta}_1 - x\lambda_2^2. \tag{B.6}$$

We will not make all intermediate results explicit, such as giving the full expressions for $A_i$ and $B_i$, which are functions of external momentum $p$, the spectral parameter $\lambda_1$ as well as the Feynman parameter $x$. Ultimately, the complete final result will be stated explicitly.

The momentum integrals are now readily solved via the standard integration formulation,

$$\int \frac{d^d q}{(2\pi)^d}\frac{q^{2m}}{(q^2+\Delta)^n} = \frac{1}{(4\pi)^{d/2}}\frac{\Gamma(m+\frac{d}{2})\Gamma(n-\frac{d}{2}-m)}{\Gamma(\frac{d}{2})\Gamma(n)}\Delta^{m+d/2-n}, \tag{B.7}$$

with $m$ a non-negative and $n$ a positive integer.

## B.2 Feynman parameter integration

Reordering the expression in powers of the Feynman parameter $x$ and taking the limit $d \to 4-2\varepsilon$, we arrive at

$$I(p,\lambda_1,\lambda_2) = \Big(\frac{1}{\varepsilon} + \log\frac{4\pi\mu^2}{e^{\gamma_E}}\Big)\sum_{i=0}^3 \frac{\alpha_i^{(f)} - \alpha_i^{(g)}}{i+1} - \int_x \sum_{i=0}^3 x^i\big(\alpha_i^{(f)}\log\tilde{\Delta}_1 - \alpha_i^{(g)}\log\tilde{\Delta}_2\big) + \mathcal{O}(\varepsilon), \tag{B.8}$$

with $\gamma_E$ the Euler-Mascheroni constant. The coefficients $\alpha_i$ and $\beta_i$ do not depend on $x$, and will be given down below. We can solve the Feynman parameter integrals analytically and simplify the first sum to obtain the final result,

$$I(p,\lambda_1,\lambda_2) = \Big(\frac{1}{\varepsilon} + \log\frac{4\pi\mu^2}{e^{\gamma_E}}\Big)\frac{3}{4}p^2 - \sum_{i=0}^3 \big[\alpha_i^{(f)} f_i - \alpha_i^{(g)} g_i\big]. \tag{B.9}$$

The coefficients $\alpha_i^{(f,g)}$ are defined as follows:

$$\alpha_0^{(f)} = \frac{p^2}{2},$$

$$\alpha_1^{(f)} = -\frac{p^2(p^2 - 5\lambda_1^2 + \lambda_2^2)}{2\lambda_1^2},$$

$$\alpha_2^{(f)} = \frac{3p^2(3p^2 - 2\lambda_1^2 + 2\lambda_2^2)}{2\lambda_1^2},$$

$$\alpha_3^{(f)} = -\frac{4p^4}{\lambda_1^2}, \tag{B.10}$$

and

$$\alpha_0^{(g)} = 0,$$

$$\alpha_1^{(g)} = -\frac{p^2(p^2 + \lambda_2^2)}{2\lambda_1^2},$$

$$\alpha_2^{(g)} = \frac{3p^2(3p^2 + 2\lambda_2^2)}{2\lambda_1^2},$$

$$\alpha_3^{(g)} = -\frac{4p^4}{\lambda_1^2}. \tag{B.11}$$

The functions $f_i$ and $g_i$ carry the branch cuts ultimately giving rise to the spectral function and are defined by integrals over the Feynman parameter $x$ via

$$f_i = \int_0^1 dx\, x^i \log \tilde{\Delta}_1, \qquad g_i = \int_0^1 dx\, x^i \log \tilde{\Delta}_2, \tag{B.12}$$

yielding

$$f_0 = \frac{\zeta}{2p^2} D_{\text{cut}} + 2\log\lambda_2 + \frac{p^2 - \lambda_1^2 + \lambda_2^2}{p^2} \log\left(\frac{\lambda_1}{\lambda_2}\right) - 2,$$

$$f_1 = \frac{1}{4p^4\zeta} D_{\text{cut}}\left[\left((\lambda_1 - \lambda_2)^2 + p^2\right)\left((\lambda_1 + \lambda_2)^2 + p^2\right)\left(p^2 - \lambda_1^2 + \lambda_2^2\right)\right] + \log\lambda_2 - \frac{p^2 - \lambda_1^2 + \lambda_2^2}{2p^2}$$
$$+ \frac{(\lambda_1^2 - \lambda_2^2)^2 + 2\lambda_2^2 p^2 + p^4}{2p^4} \log\left(\frac{\lambda_1}{\lambda_2}\right) - \frac{1}{2},$$

$$f_2 = \frac{1}{6p^6\zeta} D_{\text{cut}}\left[(\lambda_1^2 - \lambda_2^2)^4 + p^6(\lambda_1^2 + 4\lambda_2^2) - 2\lambda_2^2 p^4(\lambda_1^2 - 3\lambda_2^2)\right.$$
$$\left. + p^2(\lambda_1^2 - \lambda_2^2)^2(\lambda_1^2 + 4\lambda_2^2) + p^8\right]$$
$$+ \frac{1}{3}\log\lambda_2^2 - \frac{\lambda_2^2 - \lambda_1^2 + p^2}{6p^2} - \frac{(\lambda_1^2 - \lambda_2^2)^2 + 2\lambda_2^2 p^2 + p^4}{3p^4}$$
$$+ \frac{3\lambda_2^2 p^2(\lambda_2^2 - \lambda_1^2 + p^2) - (\lambda_1^2 - \lambda_2^2)^3 + p^6}{3p^6} \log\left(\frac{\lambda_1}{\lambda_2}\right) - \frac{2}{9},$$

$$f_3 = \frac{1}{8p^8\zeta}D_{\text{cut}}\Big[\big((\lambda_1-\lambda_2)^2+p^2\big)\big(\lambda_2^2-\lambda_1^2+p^2\big)$$

$$\times\big(\lambda_1^4+\lambda_2^4+p^4+2\lambda_2^2(p^2-\lambda_1^2)\big)\big((\lambda_1+\lambda_2)^2+p^2\big)\Big]$$

$$-\frac{1}{8}\log(-\lambda_2^2)+\frac{1}{4}\log(\lambda_2^2)+\frac{\lambda_1^2-13\lambda_2^2}{12p^2}-\frac{\lambda_1^4-8\lambda_1^2\lambda_2^2+7\lambda_2^4}{8p^4}+\frac{(\lambda_1^2-\lambda_2^2)^3}{4p^6}-\frac{7}{12}$$

$$+\frac{\log(-\lambda_1^2)}{8p^8}\Big[(\lambda_1^2-\lambda_2^2)^4+p^4\big(6\lambda_2^4-4\lambda_1^2\lambda_2^2\big)+4\lambda_2^2p^2(\lambda_1^2-\lambda_2^2)^2+4\lambda_2^2p^6+p^8\Big]$$

$$-\frac{\log(-\lambda_2^2)}{8p^8}\Big[\lambda_1^8+\lambda_2^8+4\lambda_2^6(p^2-\lambda_1^2)+2\lambda_2^4\big(3\lambda_1^4-4\lambda_1^2p^2+3p^4\big)$$

$$+4\lambda_2^2(p^2-\lambda_1^2)(\lambda_1^4+p^4)\Big], \tag{B.13}$$

where we defined

$$D_{\text{cut}} = \log\big(\zeta+\lambda_1^2-\lambda_2^2+p^2\big)-\log\big(\zeta+\lambda_1^2-\lambda_2^2-p^2\big)$$
$$+\log\big(\zeta-\lambda_1^2+\lambda_2^2+p^2\big)-\log\big(\zeta-\lambda_1^2+\lambda_2^2-p^2\big), \tag{B.14}$$

with $\zeta=\sqrt{\lambda_2^4+(\lambda_1^2+p^2)^2+2\lambda_2^2(p^2-\lambda_1^2)}$, and

$$g_0 = \log\lambda_2^2-\frac{(\lambda_2^2+p^2)\log-\lambda_2^2}{p^2}+\Big(\frac{\lambda_2^2}{p^2}+1\Big)\log\big(-\lambda_2^2-p^2\big)-2, \tag{B.15}$$

$$g_1 = \frac{1}{2p^4}\Big[-p^2(\lambda_2^2+2p^2)+p^4\log\lambda_2^2-\log-\lambda_2^2(\lambda_2^2+p^2)^2+(\lambda_2^2+p^2)^2\log\big(-\lambda_2^2-p^2\big)\Big],$$

$$g_2 = -\frac{1}{18p^6}\Big[15\lambda_2^2p^4+6\lambda_2^4p^2-6p^6\log\lambda_2^2+6\log-\lambda_2^2](\lambda_2^2+p^2)^3$$

$$-6(\lambda_2^2+p^2)^3\log\big(-\lambda_2^2-p^2\big)+13p^6\Big],$$

$$g_3 = \frac{1}{24p^8}\Big[-p^2\big(6\lambda_2^6+26\lambda_2^2p^4+21\lambda_2^4p^2+14p^6\big)+6p^8\log\lambda_2^2$$

$$-6\log-\lambda_2^2(\lambda_2^2+p^2)^4+6(\lambda_2^2+p^2)^4\log\big(-\lambda_2^2-p^2\big)\Big].$$

The gluon and ghost loops $\mathcal{D}_{\text{gluon}}$ and $\mathcal{D}_{\text{ghost}}$ featuring in the gluon self-energy $\Sigma_{AA}$ defined in (23) are computed analogously. As for the ghost self-energy, we first define

$$\mathcal{D}_{\text{gluon}}(p) = g^2\delta^{ab}C_A\int_{\lambda_1,\lambda_2}\rho_A(\lambda_2)\rho_A(\lambda_2)I_{\text{glu}}(p,\lambda_1,\lambda_2), \tag{B.16}$$

$$\mathcal{D}_{\text{ghost}}(p) = g^2\delta^{ab}C_A\int_{\lambda_1,\lambda_2}\rho_c(\lambda_2)\rho_c(\lambda_2)I_{\text{ghost}}(p,\lambda_1,\lambda_2). \tag{B.17}$$

We just quote the results for the momentum integrals $I_{\text{glu}}$ and $I_{\text{ghost}}$ as

$$I_{\text{glu}}(p,\lambda_1,\lambda_2) = \Big(\frac{1}{\varepsilon}+\log\frac{4\pi\mu^2}{e^{\gamma_E}}\Big)\Big[\frac{25}{12}p^2+\frac{3}{2}(\lambda_1^2+\lambda_2^2)\Big]$$

$$-\sum_{i=0}^{4}\big(\beta_i^{(\text{f})}f_i+\beta_i^{(\text{h})}h_i-\beta_i^{(\text{g})}g_i-\beta_i^{(\text{j})}j_i\big), \tag{B.18}$$

$$I_{\text{ghost}}(p,\lambda_1,\lambda_2) = \Big(\frac{1}{\varepsilon}+\log\frac{4\pi\mu^2}{e^{\gamma_E}}\Big)\Big[\frac{1}{12}p^2+\frac{1}{4}(\lambda_1^2+\lambda_2^2)\Big]-\mathcal{F}_{\text{ghost}}.$$

The coefficients $\beta_i^{(\cdot)}$ are defined as

$$\beta_0^{(f)} = \frac{27(\lambda_1^4 - 3\lambda_1^2\lambda_2^2) + 6p^4 + 6p^2(6\lambda_1^2 + 5\lambda_2^2)}{8\lambda_2^2}, \tag{B.19}$$

$$\beta_1^{(f)} = -\frac{3}{4\lambda_1^2\lambda_2^2}\Big[9(\lambda_1^3 - \lambda_1\lambda_2^2)^2 + p^6 + p^4(6\lambda_2^2 - 7\lambda_1^2)$$
$$+ p^2(5\lambda_2^4 - 3\lambda_1^4 - 20\lambda_1^2\lambda_2^2)\Big],$$

$$\beta_2^{(f)} = \frac{3}{8\lambda_1^2\lambda_2^2}\Big[2p^6 + 9(\lambda_1^2 - \lambda_2^2)^2(\lambda_1^2 + \lambda_2^2)$$
$$+ p^4(79\lambda_2^2 - 11\lambda_1^2) + p^2(62\lambda_2^4 - 58\lambda_1^4 - 40\lambda_1^2\lambda_2^2)\Big],$$

$$\beta_3^{(f)} = -\frac{15p^2\Big[-2\lambda_1^4 + 2\lambda_2^4 + p^2(2\lambda_1^2 + 5\lambda_2^2)\Big]}{2\lambda_1^2\lambda_2^2},$$

$$\beta_4^{(f)} = \frac{105p^4(\lambda_1^2 + \lambda_2^2)}{8\lambda_1^2\lambda_2^2},$$

$$\beta_0^{(g)} = 0, \tag{B.20}$$

$$\beta_1^{(g)} = -\frac{3p^2(p^2 + \lambda_2^2)(p^2 + 5\lambda_2^2)}{4\lambda_1^2\lambda_2^2},$$

$$\beta_2^{(g)} = \frac{3(2p^6 + 79p^4\lambda_2^2 + 62p^2\lambda_2^4 + 9\lambda_2^6)}{8\lambda_1^2\lambda_2^2},$$

$$\beta_3^{(g)} = -\frac{15(5p^4 + 2p^2\lambda_2^2)}{2\lambda_1^2},$$

$$\beta_4^{(g)} = \frac{105p^4}{8\lambda_1^2},$$

$$\beta_0^{(h)} = \beta_3^{(h)} = \beta_4^{(h)} = 0, \tag{B.21}$$

$$\beta_1^{(h)} = -\beta_2^{(h)} = -\frac{3p^6}{4\lambda_1^2\lambda_2^2},$$

$$\beta_0^{(j)} = \frac{3(2p^4 + 12p^2\lambda_1^2 + 9\lambda_1^4)}{8\lambda_2^2}, \tag{B.22}$$

$$\beta_1^{(j)} = -\frac{3(p^6 - 7p^4\lambda_1^2 - 3p^2\lambda_1^4 + 9\lambda_1^6)}{4\lambda_1^2\lambda_2^2)},$$

$$\beta_2^{(j)} = \frac{3(2p^6 - 11p^4\lambda_1^2 - 58p^2\lambda_1^4 + 9\lambda_1^6)}{8\lambda_1^2\lambda_2^2},$$

$$\beta_3^{(j)} = \frac{15p^2(-p^2 + \lambda_1^2)}{\lambda_2^2},$$

$$\beta_4^{(j)} = \frac{105p^4}{8\lambda_2^2}.$$

The functions $f_i$ and $g_i$ appearing in (B.18) have already been defined in (B.13) and (B.15). The functions $h_i$ and $j_i$ are given by

$$h_0 = 2h_1 = -2 + \log p^2, \tag{B.23}$$

$$h_2 = \frac{1}{18}\left(-13 + 6\log p^2\right),$$

$$h_3 = \frac{1}{12}\left(-7 + 3\log p^2\right),$$

$$h_4 = -\frac{149}{300} + \frac{1}{5}\log p^2,$$

as well as

$$j_0 = \frac{\lambda_1^2\left[\log\left(\lambda_1^2 + p^2\right) - \log\lambda_1^2\right]}{p^2} - 2 + \log\left(\lambda_1^2 + p^2\right),$$

$$j_1 = \frac{\lambda_1^4\log\lambda_1^2 - 2p^4 + \lambda_1^2 p^2 + \left(p^4 - \lambda_1^4\right)\log\left(\lambda_1^2 + p^2\right)}{2p^4},$$

$$j_2 = \frac{1}{18p^6}\left[-13p^6 + 3p^4\lambda_1^2 - 6p^2\lambda_1^4 - 6\lambda_1^6\log\lambda_1^2 + 6\left(p^6 + \lambda_1^6\right)\log\left(p^2 + \lambda_1^2\right)\right], \tag{B.24}$$

$$j_3 = \frac{1}{24p^8}\left[-14p^8 + 2p^6\lambda_1^2 - 3p^4\lambda_1^4 + 6p^2\lambda_1^6 + 6\lambda_1^8\log\lambda_1^2 + 6\left(p^8 - \lambda_1^8\right)\log\left(p^2 + \lambda_1^2\right)\right],$$

$$j_4 = \frac{1}{300p^{10}}\Big[-149p^{10} + 15p^8\lambda_1^2 - 20p^6\lambda_1^4 + 30p^4\lambda_1^6$$

$$- 60p^2\lambda_1^8 - 60\lambda_1^{10}\log\lambda_1^2 + 60\left(p^{10} + \lambda_1^{10}\right)\log\left(p^2 + \lambda_1^2\right)\Big].$$

The function $\mathcal{F}_{\text{ghost}}$ in (B.16) is defined as

$$\mathcal{F}_{\text{ghost}} = \frac{1}{36}\Bigg(-24\left(\lambda_1^2 + \lambda_2^2\right) - \frac{6\left(\lambda_1^2 - \lambda_2^2\right)^2}{p^2} + 6\left[3\left(\lambda_1^2 + \lambda_2^2\right) + p^2\right]\log\lambda_2^2 - 10p^2 \tag{B.25}$$

$$+ \frac{3}{p^4}\Bigg\{\left[\lambda_1^4 + \lambda_1^2\left(4p^2 - 2\lambda_2^2\right) + \left(\lambda_2^2 + p^2\right)^2\right]\left(p^2 - \lambda_1^2 + \lambda_2^2\right)\left(\log\left(-\lambda_1^2\right) - \log\left(-\lambda_2^2\right)\right)$$

$$- 2i\zeta^3\left[\arctan\left(\frac{p^2 + \lambda_1^2 - \lambda_2^2}{i\zeta}\right) + \arctan\left(\frac{p^2 - \lambda_1^2 + \lambda_2^2}{i\zeta}\right)\right]\Bigg\}\Bigg).$$

## B.3 Real frequencies

For real-time expressions of the DSE diagrams, we need (B.9) and (B.18) at real frequencies $\omega$, i.e. $I(\omega, \lambda_1, \lambda_2) := I(-i(\omega + i0^+))$. From the definitions of the respective functions and coefficients, the corresponding real-time expressions are obtained by replacing $p \rightarrow -i(\omega + i\varepsilon)$ and explicitly taking the limit $\varepsilon \searrow 0$. The calculations here were performed in WOLFRAM MATHEMATICA 12.1 with the convention $\text{Im}\log x = \pi$ for $x < 0$ for the logarithmic branch cut. In this case, for the ghost self-energy (B.9) as well as $I_{\text{ghost}}$ in (B.18) taking the above limit corresponds to the mere substitution $p \rightarrow i\omega$. For $I_{\text{glu}}$ in (B.18) this is not the case due to

symbolic manipulations that have been performed in order to simplify the expressions. Here, appropriate imaginary parts need to be added in order to get the correct limit when explicitly taking the limit $\varepsilon \searrow 0$. Note that the manual addition of appropriate imaginary parts might also be necessary for other branch cut conventions.

## B.4 Complex frequencies and spectral masses

The non-trivial analytic solutions of the Feynman parameter integrals in this work (Appendix B.2), such as (B.8), always require numerical cross-check. Especially for arbitrary complex spectral values and frequencies $\lambda_{1/2}^2, p^2 \in \mathbb{C}$, this is crucial. This becomes clear when considering the in Appendix B presented solutions for the loop momentum integrals of the diagrams in this work. For $\lambda_{1/2}^2, p^2 \in \mathbb{C}$, (B.9) and (B.18) generally do not need to hold. We will discuss this at the example of the calculation presented in Appendix B. While, after introduction of Feynman parameters (B.4), the solution of momentum integration (B.7) is still valid for $\lambda_{1/2}^2, p^2 \in \mathbb{C}$, this is generally not true for the analytic solution of the Feynman parameter integral in (B.8). For the diagrams involved in this work, the (non-trivial) Feynman parameter integrals takes the general form

$$J_{\mathrm{FP}}^i(p) = \int_0^1 \mathrm{d}x \, x^i \log\left((1-x)\lambda_1^2 + x\lambda_2^2 + x(1-x)p^2\right). \tag{B.26}$$

For certain combinations $\lambda_{1/2}^2, p^2$, the integration contour in (B.26), which is the straight line connecting 0 and 1, is now crossing the logarithmic branch of the integrand. To study the case of a pair of complex conjugate poles, the case $\lambda_2 = \bar{\lambda}_1$ is of particular interest. There, for $p^2 \leq 2\,\mathrm{Re}\,\lambda_1^2$, the integration contour always crosses the branch cut. In this case, the Feynman parameter integral in (B.26) becomes ill-defined. The reason for that lies in the introduction of Feynman parameters in the first place. The Feynman trick (B.4) is only valid if the straight line connecting $A$ and $B$ does not cross the origin, i.e. the RHS of (B.4) has no (non-integrable) pole in the integration contour. For the above described case of $\lambda_2 = \bar{\lambda}_1$ and $p^2 < 2\,\mathrm{Re}\lambda_1^2$, this is exactly what happens, however. After a shift in the loop momentum, the order of the momentum and Feynman parameter integration are interchanged. For $\lambda_2 = \bar{\lambda}_1$ and $p^2 < 2\,\mathrm{Re}\lambda_1^2$, there always exists a value of the loop momentum $q$ for which the Feynman parameter integration contour crosses a non-integrable pole. Since the $q$ integration is performed first, this pole manifests itself as a branch cut in the Feynman parameter integration. The Feynman parameter integral becomes ill-defined, since the Feynman trick (B.4) is not well-defined in the first place in this case and can not be used to solve the momentum integral in this case.

For certain combinations of $\lambda_{1/2}^2, p^2 \in \mathbb{C}$ the branch cuts resulting from poles in the Feynman parameter integration domain can be avoided by contour deformation for the Feynman parameter integral. The Feynman parameter $x$ is then integrated between 0 and 1 along an arbitrary curve in the complex plane which avoids the branch cut(s). In that case, a numeric solution of the Feynman parameter in can be well treated numerically along with possible spectral integrals. In Section 4, we apply the described contour deformation to verify the analytic solutions for the Feynman parameter integrals. The development of a systematic procedure for finding contours avoiding these branch cuts is deferred to the future.

A possible other approach to tackle the momentum integral for arbitrary complex spectral parameters and frequencies lies in the Mellin-Barnes representation of propagators (E.5), which also holds for complex masses. In Appendix E, we utilise this representation to calculate the ghost loop of the gluon DSE in a particular parametrisation of the ghost propagator (E.3) involving a massive non-integer propagator power part.

# C Integral representation for propagators with multiple branch cuts

The analytic structure of a propagator $G$ obeying the KL representation (7) is tightly constrained by the nature of the former integral representation. The necessary conditions for the spectral representation to exist include

(i) *Holomorphicity:* $G$ is holomorphic in the upper half plane $\mathbb{H} = \{z \,|\, \mathrm{Im}\, z > 0\}$,

(ii) *Mirror symmetry:* $G(z) = \bar{G}(\bar{z})$ and $\mathrm{Im}\, G(z) = 0$ for $z > 0$,

(iii) *Asymptotic decay:* $|z\, G(z)| \to 0$ for $|z| \to \infty$,

(iv) *Spectral convergence:*

    (IR)   $|z\, G(z)| < \infty$ for $z \to 0$,

    (UV)   $|\log z \,\mathrm{Im}\, G(z)| \to 0$ for $z \to -\infty$.

Items (i) and (ii) roughly speaking guarantee that the spectral kernel has the form $1/(p^2 + \lambda^2)$ and the spectral function is defined via (8) with the integration domain restricted to $\lambda^2 > 0$ by (iii). (iv) guarantees for convergence of the spectral integral.

**Ordinary Källén-Lehmann representation**   With properties (i-iv), the spectral representation can be derived explicitly via Cauchy's integral formula. It states for a holomorphic function $G$ defined an open set $U \subset \mathbb{C}$, $G : U \to \mathbb{C}$, that the value of $G$ at any point $z_0$ enclosed by an arbitrary, closed rectifiable curve $\gamma$ in $U$ is given by

$$G(z_0) = \frac{1}{2\pi i} \oint_\gamma dz\, \frac{G(z)}{z - z_0}\,. \tag{C.1}$$

We want to find $\gamma$ such that we can use (C.1) for all $z_0 \in \mathbb{C}$, for which the easiest choice would be the circle around the origin $\mathcal{C}_R$ and taking $R \to \infty$. Since for $\mathrm{Im}\, G(z) \neq 0$ for $z < 0$, $G$ is discontinuous along the negative real axis according to (ii) however, we explicitly need to exclude this region from the integration contour by going from negative infinity towards the origin along just above the negative real axis, turning at the origin and then returning to negative infinity along below the real axis. We can then recast (C.1) as

$$G(z_0) = \frac{1}{2\pi i} \lim_{R \to \infty} \left( \int_{\mathcal{C}_R} dz\, \frac{G(z)}{z - z_0} - \int_0^R dz\, \frac{G(-z + i\varepsilon)}{z + z_0 - i\varepsilon} + \int_0^R dz\, \frac{G(-z - i\varepsilon)}{z + z_0 + i\varepsilon} \right), \tag{C.2}$$

where in the last two terms the integration boundaries have been interchanged, and we substituted $z \to -z$. Due to (iii), the first term vanishes according to Jordan's Lemma. Exploiting the mirror symmetry (ii), we can combine the latter two terms, since their real parts cancel. We find that

$$G(z_0) = \int_0^\infty \frac{dz}{2\pi} \frac{2\, \mathrm{Im}\, G(-z - i\varepsilon)}{z + z_0}\,, \tag{C.3}$$

which is the well-known Källén-Lehmann representation. Note that formally, $G$ receives another contribution in the limit $\varepsilon \to 0$ due to the opposite signs of $\varepsilon$ in the denominators in the last two terms of (C.2), which is

$$-\int_0^\infty \frac{dz}{\pi} \mathrm{Re}\, G(-z - i\varepsilon) \frac{\varepsilon}{(z + z_0)^2 + \varepsilon^2}\,. \tag{C.4}$$

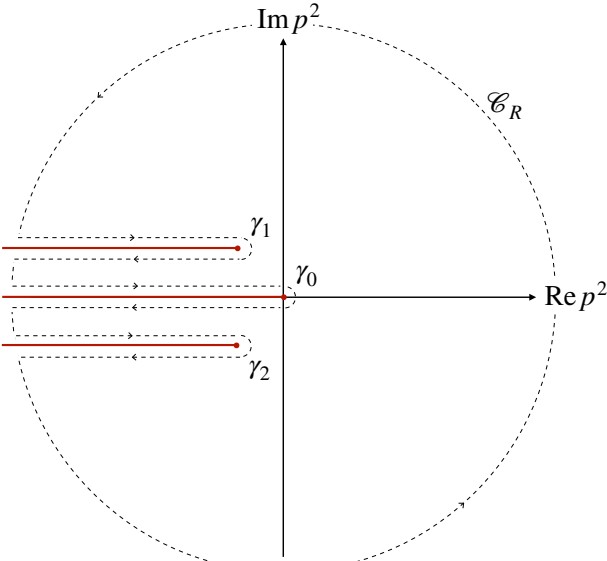

Figure 10: Integration contour $\gamma$ in Cauchy's theorem (C.1) for construction of an integral representation of the ghost propagator from Appendix D.1, which uses an input gluon propagator with complex conjugate poles. Branch cuts are marked with red lines. The ghost propagator shows the ordinary branch cut along the negative real axis and two additional branch cuts, starting at $\chi$ and $\bar{\chi}$ and stretching in parallel to the real axis towards negative infinity, comp. also Figure 11. All branch cuts are explicitly excluded from the integration contour by the $\gamma_i$'s.

Generally, $\lim_{\varepsilon \to 0} \varepsilon/((z+z_0)^2 + \varepsilon^2)$ is a representation of the delta distribution $\delta(z+z_0)$. Here however, for $\varepsilon \to 0$ this term vanishes since $z = -z_0$ is not contained in the integration domain. By definition of Cauchy's formula, $z_0$ lies inside $\gamma$, while the integration variable $-z \in \mathbb{R}^-$, which is defined on the branch cut, does not.

**Propagators with multiple branch cuts**   In the case of a gluon propagator with complex conjugate poles as considered in Appendix D.1, the ghost propagator shows two additional branch cuts, see Figure 11. These additional cuts start at $p^2 = -\chi^2$ and $-\bar{\chi}^2$ respectively and stretch parallel to the real axis towards negative infinity. This general integral representation for propagators with multiple branch cuts $\mathcal{B}_i$ can now be constructed in analogy to (C.1)-(C.3). By the existence of additional branch cuts we need to relax property (i), still assuming holomorphicity everywhere except for the cuts, however. As for the derivation of the KL representation above, this is done by choosing the integration contour to wind around the cuts by simply excluding these additional branch cuts from the integration contour $\gamma$. We go from the cuts asymptotic limit to the branch point $\chi_i$ infinitesimally above/below the cut, turning at the branch point at returning the same path just infinitesimally below/above the cut. For the case of the ghost propagator of Appendix D.1 which has three branch cuts, the integration contour is displayed in Figure 10.

The full integration contour can then conveniently be written as

$$\gamma = \lim_{R \to \infty} \mathcal{C}_R \bigoplus_i \gamma_i \, , \tag{C.5}$$

where $\gamma_0$ is the contour winding around the usual branch cut of the KL representation along the negative real axis. As before, due to property (iii), the integration along $\mathcal{C}_R$ vanishes.

From (C.1), by above choice of $\gamma$ we arrive at

$$G(z_0) = \frac{1}{2\pi i} \sum_i \int_{\gamma_i} dz \frac{G(z)}{z - z_0} \,. \tag{C.6}$$

We now split $\gamma_i$ into the parts above/below the cut, which we call $\mathcal{B}_i^{+/-} = \mathcal{B}_i \pm i\varepsilon$, such that $\gamma_i = \mathcal{B}_i^+ \oplus \mathcal{B}_i^-$. Since we integrate along the path in the mathematically positive direction, if the asymptotic value at infinity of the cut $\mathcal{B}_i$, $B_i^\infty$, lies in the left half plane, $\gamma_i$ starts above the cut with $\mathcal{B}_i^+$. The direction of integration is then such that we integrate along $\mathcal{B}_i^+$ from $B_i^\infty + i\varepsilon$ to $\chi_i + i\varepsilon$, and then go back along $\mathcal{B}_i^-$ from $\chi_i - i\varepsilon$ to $B_i^\infty - i\varepsilon$. If the asymptotic value lies in the right half plane, this works vice versa, going along $\mathcal{B}_i^-$ from $B_i^\infty - i\varepsilon$ to $\chi_i - i\varepsilon$ first and then back. Plugging in the split of $\gamma_i$ explicitly and assuming the appropriate directionality along $\mathcal{B}_i^{+/-}$, we arrive at

$$G(z_0) = \frac{1}{2\pi i} \sum_i \int_{\mathcal{B}_i^+ \oplus \mathcal{B}_i^+} dz \frac{G(z)}{z - z_0} \tag{C.7}$$

$$= \frac{1}{2\pi i} \sum_i \int_{\mathcal{B}_i} dz \frac{G(z + i\varepsilon) - G(z - i\varepsilon)}{z - z_0} \,,$$

where in the second line we used that we integrate along $\mathcal{B}_i^{+/-}$ in opposite directions.

With the general integral representation (C.7) for propagators with multiple branch cuts at hand, we can now directly arrive at the modified spectral representation for the ghost propagator (D.6). With the complex structure as shown in Figure 11, the corresponding integration contour $\gamma$ is sketched in Figure 10. As demonstrated in (C.2) and (C.3), the branch cut $\mathcal{B}_0$ just yields the usual KL part $G_c^{KL}$. $\mathcal{B}_1$ and $\mathcal{B}_2$ then constitute the modification of the ordinary spectral representation, explicitly given by

$$G_c^\chi(z_0) = \frac{1}{2\pi i} \int_{\mathcal{B}_1 \oplus \mathcal{B}_2} dz \frac{G_c(z + i\varepsilon) - G_c(z - i\varepsilon)}{z - z_0} \tag{C.8}$$

$$= \frac{-1}{2\pi i} \left( \int_{-\chi^2}^{-\infty - \chi^2} dz \frac{G_c(z + i\varepsilon) - G_c(z - i\varepsilon)}{z - z_0} + \int_{-\bar{\chi}^2}^{-\infty - \bar{\chi}^2} dz \frac{G_c(z + i\varepsilon) - G_c(z - i\varepsilon)}{z - z_0} \right)$$

$$= \frac{1}{2\pi i} \int_0^\infty dz \left( \frac{G_c(-z - \chi^2 - i\varepsilon) - G_c(-z - \chi^2 + i\varepsilon)}{z + \chi^2 + z_0} \right.$$

$$\left. + \frac{G_c(-z - \bar{\chi}^2 - i\varepsilon) - G_c(-z - \bar{\chi}^2 + i\varepsilon)}{z + \bar{\chi}^2 + z_0} \right).$$

Note that in (C.8), we already dropped the contributions corresponding to (C.4) here when combining the dominators with different signs of $\varepsilon$. We can now use that $G_c$ is only discontinuous in its imaginary part across the branch cuts $\mathcal{B}_1$ and $\mathcal{B}_2$, such that, as for the KL branch cut, the real parts in the propagator difference in the denominators of (C.8) cancel. We find that

$$G_c^\chi(z_0) = \frac{1}{2\pi} \int_0^\infty dz \left( \frac{1}{z + \chi^2 + z_0} 2 \operatorname{Im}\left[ G_c(-z - \chi^2 + i\varepsilon) - G_c(-z - \chi^2 - i\varepsilon) \right] \right. \tag{C.9}$$

$$\left. + \frac{2 \operatorname{Im}\left[ G_c(-z - \bar{\chi}^2 + i\varepsilon) - G_c(-z - \bar{\chi}^2 - i\varepsilon) \right]}{z + \bar{\chi}^2 + z_0} \right).$$

Exploiting the mirror symmetry (ii), we finally arrive at

$$G_c^\chi(z_0) = \frac{1}{2\pi} \int_0^\infty dz \left( \frac{1}{z + \chi^2 + z_0} + \frac{1}{z + \bar\chi^2 + z_0} \right) \left( 2\,\mathrm{Im}\big[ G_c(-z - \chi^2 + i\varepsilon) - G_c(-z - \chi^2 - i\varepsilon) \big] \right).$$
(C.10)

With $G_c = G_c^{\mathrm{KL}} + G_c^\chi$, we end up with the modified spectral representation for the ghost propagator, which is

$$G_c(z_0) = \int_0^\infty \frac{dz}{2\pi} \left[ \frac{\rho_c^{\mathrm{KL}}(z)}{p^2 + z^2} + \rho_c^\chi(z) \left( \frac{1}{z + \chi^2 + z_0} + \frac{1}{z + \bar\chi^2 + z_0} \right) \right],$$
(C.11)

with

$$\rho_c^\chi(z) = 2\,\mathrm{Im}\big[ G_c(-z - \chi^2 + i\varepsilon) - G_c(-z - \chi^2 - i\varepsilon) \big],$$

and $\rho_c^{\mathrm{KL}}$ the usual KL spectral function (8).

# D  Propagation of non-analyticities through the coupled YM system

## D.1  Ghost DSE

As a starting point of the following investigation, we study the effect of a single pair of complex conjugate poles in the gluon propagator on the ghost propagator. This is done via the spectral ghost DSE, set up in Section 3. Owing to the spectral-non-spectral split (34), for the complex conjugate pole contribution to the gluon propagator we then explicitly have

$$G_A^\chi(p) = \frac{R_\chi}{p^2 + \chi^2} + \frac{\bar R_\chi}{p^2 + \bar\chi^2},$$
(D.1)

where one of the poles is located at $p^2 = -\chi^2$ and has residue $R_\chi$. The relevant correction to the fully spectral part of the ghost loop $\sim G_A^{\mathrm{KL}} G_c^{\mathrm{KL}}$ is then $\sim G_A^\chi G_c^{\mathrm{KL}}$. We assume the ghost propagator to be given solely by its classical contribution, i.e.

$$G_c^{\mathrm{KL}} \approx G_c^{\mathrm{cl}}, \qquad \text{with} \qquad G_c^{\mathrm{cl}}(p) = \frac{1}{p^2}.$$
(D.2)

For the ghost spectral function, this corresponds to just having the massless pole with residue $1/Z_c = 1$ in the origin, cf. (9). Note that the results of the following discussion are not altered by also including scattering tails for ghost and gluon spectral functions due to superposition with the contributions of (D.2). For the same reason, the following investigation is independent of particular infrared scenarios of Yang-Mills such as scaling/decoupling or massive solutions.

With choice (D.2) and the complex conjugate pole gluon propagator (D.1), we arrive at the ghost self-energy

$$\Sigma_{\bar c c}^{(1)}(p) = g^2 N_c \int_q \left( p^2 - \frac{(p \cdot q)^2}{q^2} \right) G_c^{\mathrm{cl}}(p+q) G_A^\chi(q)$$

$$= g^2 N_c \int_q \left( p^2 - \frac{(p \cdot q)^2}{q^2} \right) \frac{1}{(p+q)^2} \left( \frac{R_\chi}{p^2 + \chi^2} + \frac{\bar R_\chi}{p^2 + \bar\chi^2} \right),$$
(D.3)

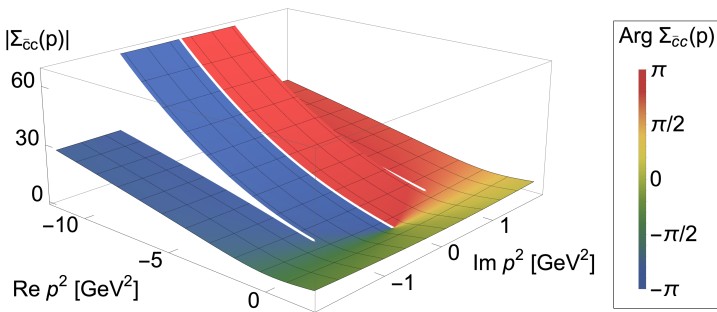

Figure 11: Ghost self energy $\Sigma_{\bar{c}c}$ in the complex plane as defined in (D.3), yielding two additional branch cuts in the ghost propagator. In $\Sigma_{\bar{c}c}$, a gluon propagator with a pair of complex conjugate poles (D.1) is used. This choice directly results in two additional branch cuts, running parallelly to the negative real axis. Hence, we observe a violation of the ghost propagators Källén-Lehmann representation induced by a pair of complex conjugate poles in the gluon propagator.

which is readily integrated analytically via dimensional regularisation in analogy to Appendix B with the appropriate choice of the gluon and ghost spectral functions. The respective gluon and ghost spectral functions of the propagators (D.1) and (D.2) read,

$$\rho_A(\omega) = \rho_A^\chi(\omega),$$
$$\rho_c(\omega) = \rho_c^{\mathrm{cl}}(\omega), \tag{D.4a}$$

with

$$\rho_A^\chi(\omega) = \pi\big[Z_\chi \delta(\omega^2 - \chi^2) + \bar{Z}_\chi \delta(\omega^2 - \bar{\chi}^2)\big],$$
$$\rho_c^{\mathrm{cl}} = \pi\delta(\omega^2). \tag{D.4b}$$

The $\delta$-distributions for complex arguments $\chi^2, \bar{\chi}^2 \in \mathbb{C}$ in the gluon spectral functions should then be understood as

$$\int_0^\infty \mathrm{d}\omega\, \delta(\omega - \chi)\Phi(\omega) = \Phi(\chi), \tag{D.5}$$

for a test function $\Phi(\omega)$. Evidently, the complex frequencies $\chi, \bar{\chi}$ are not inside the spectral integration domain $\omega \in [0, \infty)$. In order to make sense in a distributional sense, a proper integration contour for the spectral integration has to be chosen, since the complex conjugate pole positions are not element of the usual spectral integration domain, for details see Appendix C.

The analytic result for the ghost self-energy (D.3) is depicted in the full complex $p^2$-plane in Figure 11. In addition to the usual branch cut along the negative $p^2$-axis, two additional branch cuts are present, and are clearly visible in Figure 11. Starting at their respective branch points at $\chi$ and $\bar{\chi}$, the additional cuts extend parallelly to the negative real axis towards infinity. In consequence, the KL representation is violated, since it requires all non-analyticities to be confined to the negative real $p^2$-axis.

In the absence of a KL spectral representation one can devise an alternative integral representation for the ghost propagator. This representation will maintain the analytical solvability of loop momentum integrals featuring ghost propagators despite violation of its spectral representation. In consequence, also in a scenario like shown in Figure 11 functional equations can still be evaluated on the real frequency axis. Given the complete complex structure of $\Sigma_{\bar{c}c}$, this

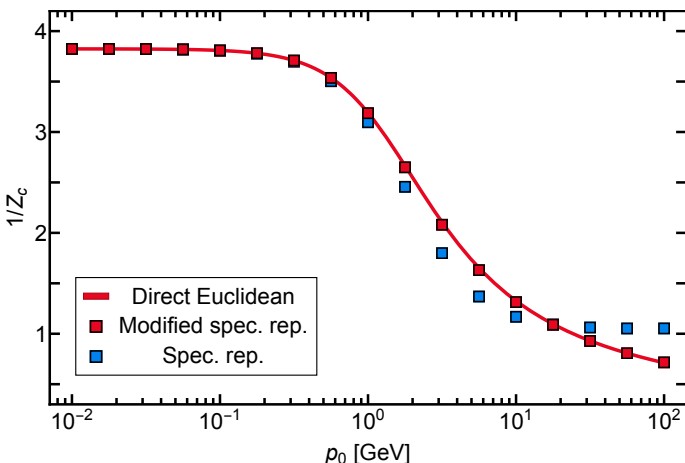

Figure 12: Violation of the ghost propagators Källén-Lehmann representation by using a gluon propagator featuring a pair of complex conjugate poles and validation of the ghost propagators modified spectral representation (D.6). The solid line represents the ghost dressing function computed directly via the spectral Euclidean DSE (29), using a complex conjugate pole gluon propagator. The squared points are obtained from the corresponding real-time DSE via the ordinary spectral representation (7) (blue) and the modified spectral representation (D.6), also taking into account the two additional branch cuts (comp. Figure 11) induced by the gluon propagators complex conjugate poles. While the dressing function obtained from the modified spectral representation matches the directly computed Euclidean ghost dressing perfectly, the Källén-Lehmann one is clearly off. Note that this not only proves the violation of the ordinary spectral representation, but in particular validates the result for the ghost self-energy (D.3) presented in Figure 11.

can be done in analogy to the construction of the KL representation (7) by help of Cauchy's integral theorem. We end up with a *modified spectral representation* for the ghost propagator by excluding also the two additional branch cuts from the circular integration contour with radius $R \to \infty$ around the origin. In the spectral-non-spectral split (34), this leads us to a non-spectral contribution of the ghost propagator given by

$$G_c^{\chi}(p) = \int_{\lambda} \rho_c^{\chi}(\lambda) \left( \frac{1}{p^2 + \lambda^2 + \chi^2} + \frac{1}{p^2 + \lambda^2 + \bar{\chi}^2} \right). \tag{D.6a}$$

We also introduced the additional spectral function $\rho_c^{\chi}$ defined via

$$\rho_c^{\chi}(\omega) = \mathrm{Im}\left[ G(-\mathrm{i}\sqrt{\omega^2 + \chi^2 + \mathrm{i}0^+}) - G(-\mathrm{i}\sqrt{\omega^2 + \chi^2 - \mathrm{i}0^+}) \right]. \tag{D.6b}$$

Note that in the Källén-Lehmann case, the imaginary parts of the two propagators in (D.6b) are related by mirror (anti)symmetry. Here, this symmetry is spoiled by the fact the branch cuts are shifted into the complex plane through the appearance of the complex mass parameter $\chi$. The spectral functions encoding the weight of the branch cuts in the upper and lower half are related by this exact mirror symmetry, however. This symmetry has been exploited in obtaining (D.6), since there only one spectral function appears. The full derivation of the modified spectral representation (D.6) as well as its generalisation to an arbitrary number of branch cuts is presented in Appendix C.

In Figure 12, we compare the directly computed Euclidean ghost propagator corresponding to the ghost self-energy defined in (D.3) with its KL as well as its modified spectral representation. The violation of the KL representation by the complex conjugate poles of the gluon

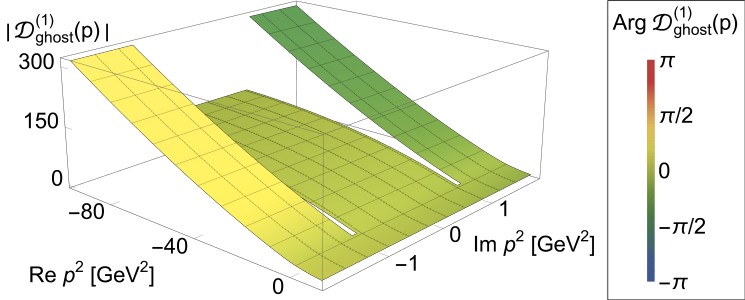

Figure 13: Contribution to the ghost loop $\mathcal{D}_{\text{ghost}}$ in the complex plane as defined in (D.7), causing additional branch cuts in the gluon propagator. In $\mathcal{D}_{\text{ghost}}^{(1)}$, one of the ghost propagators is given by the violation of the spectral representation $G_c^{\chi}$ of the ghost propagator, as defined in (D.6). The modification of the ordinary spectral representation is constituted by two additional branch cuts in the ghost propagator (comp.Figure 11), which are themselves induced by a pair of complex conjugate poles in the gluon propagator through the ghost DSE (29). In consequence, a consistent solution of Yang-Mills theory with one or more pairs of complex conjugate poles in the gluon propagator (on top of the usual branch cut) is ruled out, since we were able to show that a pair of complex conjugate poles always produces an additional, corresponding pair of branch cuts.

propagator is validated. In addition, the validity of the modified spectral representation (D.6) is confirmed. In particular, this confirms the analytic structure of the ghost self-energy presented in Figure 11, since the modified spectral representation is a direct consequence.

## D.2  Gluon DSE

We proceed with the analysis of the complex structure of Yang-Mills theory by investigating the back-propagation of a pair of complex conjugate poles in the gluon propagator into the spectral gluon DSE: In the ghost loop we insert the modified spectral representation (D.6) for the ghost, and investigate the contribution of the additional cuts. For a complete picture, the complex conjugate gluon propagator poles also have to be fed back via the gluon loops. The latter part will be deferred to future work, however, since the feedback of the additional cuts in the ghost propagator suffices to arrive at a conclusive picture. Nevertheless, we will provide the relevant expressions in this section. Note also that the tadpole is absorbed in the renormalisation.

**Ghost loop**

We use the spectral DSEs set-up Section 3, similarly to Appendix D.1 and concentrate on the leading order correction $G^{\text{KL}}G^{\chi}$. The computation and the analytic results are deferred to Appendix B. In the spectral gluon DSE, we now consider the modified spectral representation for the ghost, where the non-spectral part $G_c^{\chi}$ is constituted by (D.6). For the spectral part of the ghost propagator, we again only consider the classical contribution, see (D.2). This leads us to

$$
\mathcal{D}_{\text{ghost}}^{(1)} = g^2 N_c \int_q \left( q^2 - \frac{(p \cdot q)^2}{p^2} \right) G_c^{\chi}(p+q) G^{\text{KL}}(q)
$$

$$
= g^2 N_c \int_{\lambda} \rho_c^{\chi}(\lambda) \int_q \left( q^2 - \frac{(p \cdot q)^2}{p^2} \right) \frac{1}{(p+q)^2} \left( \frac{1}{q^2 + \lambda^2 + \chi^2} + \frac{1}{q^2 + \lambda^2 + \bar{\chi}^2} \right). \quad \text{(D.7)}
$$

Again, the loop momentum integral in (D.7) can be evaluated analytically via dimensional regularisation, see Appendix B.2. The result is obtained by adding to copies of the expression for the ghost loop quoted in Appendix B.2 where one spectral parameter is taken to zero and the other one is substituted such that the ordinary KL kernel is transformed into that of the modified spectral representation Equation (D.6) featuring in Equation (D.7). In explicit, this is $\lambda_1 \to 0$ and $\lambda_2 \to \sqrt{\lambda_2^2 + \chi^2}$ resp. $\sqrt{\lambda_2^2 + \bar{\chi}^2}$. The validity range of this substitution is discussed in Appendix B.4, since by above the substitutions the spectral parameters are effectively complex.

We now aim for a closed symbolic form for (D.7), which necessitates analytic access to the spectral integral. For the present purpose of studying the complex structure, it suffices to choose a well-behaved trial spectral function $\rho_c^\chi = \rho^{(\mathrm{trial})}$ with appropriate decay behaviour. Here, a convenient choice is $\rho^{(\mathrm{trial})}(\lambda) = 1/(1 + \lambda^2)$. The superficially divergent spectral integral is rendered finite via application of spectral BPHZ regularisation, see Section 3.3. We emphasise that both the procedure of spectral regularisation and the choice of $\rho^{(\mathrm{trial})}$, do not affect the complex structure of the diagram.

In the right panel of Figure 13 we show the leading order correction (D.7) in the complex momentum plane. We find two additional branch cuts, stretching in parallel to the real axis from $p^2 = -\chi^2$ and $-\bar{\chi}^2$ towards negative real infinity. Thus, a pair of complex conjugate poles in the gluon propagator also leads to additional branch cuts in the gluon propagator. This can be seen via the modified spectral representation for the ghost propagator (D.6), itself induced by the complex conjugate poles of the gluon propagator via the ghost DSE, see Appendix D.1.

At order $(G_c^\chi)^2$, the contribution to the ghost loop arising from the complex conjugate pole gluon propagator reads

$$
\begin{aligned}
\mathcal{D}_{\mathrm{ghost}}^{(2)} &= g^2 N_c \int_q \left( q^2 - \frac{(p \cdot q)^2}{p^2} \right) G_c^\chi(q) G_c^\chi(p+q) \\
&= g^2 N_c \int_{\lambda_1, \lambda_2} \rho_c^\chi(\lambda_1) \rho_c^\chi(\lambda_2) \int_q \left( q^2 - \frac{(p \cdot q)^2}{p^2} \right) \\
&\qquad\qquad \times \left( \frac{1}{(p+q)^2 + \lambda^2 + \chi^2} + \frac{1}{(p+q)^2 + \lambda^2 + \bar{\chi}^2} \right) \\
&\qquad\qquad \times \left( \frac{1}{q^2 + \lambda^2 + \chi^2} + \frac{1}{q^2 + \lambda^2 + \bar{\chi}^2} \right).
\end{aligned}
\tag{D.8}
$$

Equation (D.8) involves two spectral integrals, obstructing a fully analytic evaluation of this contribution. Inspecting the analytic structure of the integrand in comparison to the $G_c^\chi G_c^{\mathrm{KL}}$-contribution of (D.7), we see that the previously massless classical ghost propagator is replaced by the modified spectral kernel $1/(p^2 + \lambda^2 + \chi^2)$ and $1/(p^2 + \lambda^2 + \bar{\chi}^2)$. The complex structure of these integrals is dominated by the imaginary parts of the logarithmic terms, that occur after evaluating the momentum integrals via dimensional regularisation. Hence, we anticipate, that the complex structure of this contribution is similar to that of the leading order correction shown in Figure 13.

The direct investigation of this term is not performed here, as the leading order contribution already shows two additional branch cuts. The latter are already inconsistent with the assumption of a single pair of complex conjugate poles in the gluon propagator, which was the starting point of this investigation. Nonetheless, in the following we will also quote the expressions for the complex conjugate poles induced corrections to the gluon loop for the sake of completeness.

**Gluon loop**

The first order contribution in $G_A^\chi$ to the gluon loop is given by

$$
\begin{aligned}
\mathcal{D}_{\text{gluon}}^{(1)} &= g^2 N_c \int_q V(p,q) G_A^\chi(q) G_A^{\text{KL}}(p+q) \\
&= g^2 N_c \int_\lambda \rho_A^{\text{KL}}(\lambda) \int_q V(p,q) \frac{1}{(p+q)^2 + \lambda^2} \left( \frac{R_\chi}{q^2 + \chi^2} + \frac{\bar{R}_\chi}{q^2 + \bar{\chi}^2} \right),
\end{aligned}
\tag{D.9}
$$

with $V(p,q)$ as defined in (26). The $\mathcal{O}(G_A^{\chi\,2})$ contribution is given by

$$
\begin{aligned}
\mathcal{D}_{\text{gluon}}^{(2)} &= g^2 N_c \int_q V(p,q) G_A^\chi(q) G_A^\chi(p+q) \\
&= g^2 N_c \int_q V(p,q) \left( \frac{R_\chi}{q^2 + \chi^2} + \frac{\bar{R}_\chi}{q^2 + \bar{\chi}^2} \right) \left( \frac{R_\chi}{(p+q)^2 + \chi^2} + \frac{\bar{R}_\chi}{(p+q)^2 + \bar{\chi}^2} \right).
\end{aligned}
\tag{D.10}
$$

The computation of $\mathcal{D}_{\text{gluon}}^{(1)}$ and $\mathcal{D}_{\text{gluon}}^{(2)}$ in the full complex momentum plane requires the evaluation of the respective momentum integrals for two arbitrary complex masses $\chi, \bar{\chi}$ and momenta $p^2$. The analytic evaluation of this integral is significantly more challenging than with just one complex mass parameter, as for the ghost loop (D.7). In particular, the employed technique of Feynman parametrisation is not applicable in this scenario, as we discuss in Appendix B.4.

However, we have already shown in Appendix D.1, that a complex conjugate pole gluon propagator leads to additional branch cuts in the gluon propagator via the ghost loop. Thus, the input assumption of a spectral function plus a pair of complex conjugate poles for the gluon propagator is violated independently of the complex structure of the gluon loop $\mathcal{D}_{\text{gluon}}$. While an investigation of the effect of the complex conjugate pole contribution of the gluon propagator on the complex structure of the gluon loop might nevertheless yield additional valuable insight into the analytic structure of Yang-Mills theory, we defer this to future work. Still, we remark that in our opinion a cancellation between the shifted branch cuts of $\mathcal{D}_{\text{ghost}}$ and those possibly existing in $\mathcal{D}_{\text{gluon}}$ cannot be expected. This would require the vertices to compensate for the different weights of the cuts, since the ghost diagram cuts are induced by the ghost and the (possible) gluon diagram cuts by the gluon propagator.

# E  Ghost loop with massive non-integer power propagators

The scaling solution of Yang-Mills theory is characterised by the IR behaviour of the ghost and gluon propagator dressing functions as

$$
\lim_{p \to 0} Z_A(p) \sim (p^2)^{-2\kappa}, \qquad \lim_{p \to 0} Z_c(p) = (p^2)^\kappa,
\tag{E.1}
$$

while for a decoupling behaviour, we have

$$
\lim_{p \to 0} Z_A(p) \sim \frac{1}{p^2}, \qquad \lim_{p \to 0} Z_c(p) = Z_c,
\tag{E.2}
$$

A particularly useful analytic form of the ghost propagator which allows to smoothly interpolate between scaling and decoupling behaviour in the IR is given by

$$
G_c(p,m) = \frac{1}{p^2 (p+m)^\kappa},
\tag{E.3}
$$

with the non-integer scaling exponent $0 < \kappa < 1$. The scaling solution is realised for $m \to 0$. Non-perturbative studies of Yang-Mills theories suggest $\kappa \approx 0.57$ [46]. In an approximation with bare vertices, the value of $\kappa$ can be determined analytically from the DSE to be $\kappa = \frac{93 + \sqrt{1201}}{98} \approx 0.59535$ [95].

In cases like the scaling or decoupling scenario where the infrared behaviour of a propagator is known, it can be beneficial to analytically split off the IR part as $G = G^{\text{IR}} + \Delta G$. Here, we study the ghost loop $\mathcal{D}_{\text{ghost}}$ in the gluon DSE where the ghost propagator is entirely given by the IR parametrisation of (E.3), reading

$$\mathcal{D}_{\text{ghost}} = g^2 N_c \tilde{Z}_1 \int_q \left( q^2 - \frac{(p \cdot q)^2}{p^2} \right) \frac{1}{q^2} \frac{1}{(p+q)^2} \frac{1}{(q^2+m^2)^\kappa} \frac{1}{((p+q)^2+m^2)^\kappa} \, . \quad \text{(E.4)}$$

Analytic solutions of integrals of this kind have, to our knowledge, not been quoted in the literature so far. The non-integer exponent $\kappa$ increases the difficulty of the integral enormously. Since only the non-integer part of the propagator power carries the mass $m$, from the mathematical perspective (E.4) represents a Feynman diagram with four propagators in a particular momentum-configuration with two massive propagators of the same mass. The large number of propagators renders the approach of introducing Feynman parameters as in Appendix B non-feasible. A more powerful technique to solve integrals of this kind has been proposed by Davydychev and Boos [100], representing massive denominators by Mellin-Barnes integrals as

$$\frac{1}{(k^2+m^2)^\alpha} = \frac{1}{(k^2)^\alpha} \frac{1}{\Gamma(\alpha)} \frac{1}{2\pi i} \int_{-i\infty}^{i\infty} ds \left( \frac{m^2}{k^2} \right)^s \Gamma(-s)\Gamma(\alpha+s), \quad \text{(E.5)}$$

which follows from the Barnes integral representation of the hypergeometric function $_1F_0(a|z)$. A pedagogical introduction to the technique can be found e.g. in [101]. The generalised hypergeometric function of one variable is defined by

$$_A F_B \left( \begin{array}{c} a_1, \ldots, a_A \\ b_1, \ldots, b_B \end{array} \middle| z \right) = \sum_{j=0}^\infty \frac{(a_1)_j \ldots (a_A)_j}{(b_1)_j \ldots (b_B)_j} \frac{z^j}{j!}, \quad \text{(E.6)}$$

where $(a)_j = \Gamma(a+j)/\Gamma(a)$ is the Pochhammer symbol.

Using (E.5) for the non-integer power propagators in (E.4) and dropping the prefactor $g^2 N_c \tilde{Z}_1$, we get

$$\mathcal{D}_{\text{ghost}} = \frac{-1}{4\pi^2} \frac{1}{\Gamma(\kappa)^2} \int_{-i\infty}^{i\infty} ds \int_{-i\infty}^{i\infty} dt \, (m^2)^{s+t} \Gamma(-s)\Gamma(-t)\Gamma(\kappa+s)\Gamma(\kappa+t) I_{\text{ghost}}^{\text{MB}}(p,m), \quad \text{(E.7)}$$

with

$$I_{\text{ghost}}^{\text{MB}} = \int_q \left( q^2 - \frac{(p \cdot q)^2}{p^2} \right) \frac{1}{(q^2)^{\kappa+s+1}} \frac{1}{((p+q)^2)^{\kappa+t+1}} \, .$$

Defining $k = p + q$, we can rewrite the momentum integral as

$$I_{\text{ghost}}^{\text{MB}} = \frac{1}{2} \int_q \mathcal{P}(p,q,k) \frac{1}{(q^2)^{\kappa+s+1}} \frac{1}{(k^2)^{\kappa+t+1}} \, , \quad \text{(E.8)}$$

where

$$\mathcal{P}(p,q,k) = q^2 + k^2 - \frac{1}{2}p^2 - \frac{1}{2p^2}\left(k^4 - 2k^2 q^2 + q^4\right).$$

Equation (E.8) is now evaluated with the help of the well-known integration formula

$$\int \frac{d^d q}{(2\pi)^d}\Big(\frac{1}{q^2}\Big)^{d/2-\alpha}\Big(\frac{1}{p^2}\Big)^{\alpha-\beta}\Big(\frac{1}{k^2}\Big)^{\beta} = \frac{1}{(4\pi)^{d/2}}\frac{\Gamma(\alpha)\Gamma(\frac{d}{2}-\beta)\Gamma(\beta-\alpha)}{\Gamma(\beta)\Gamma(\frac{d}{2}-\alpha)\Gamma(\frac{d}{2}+\alpha-\beta)}. \tag{E.9}$$

Convergence of (E.9) is only ensured for $\mathrm{Re}(\alpha) > 0$, $\mathrm{Re}(\beta-\alpha) > 0$ and $\mathrm{Re}(\beta) < d/2$. Although the convergence requirements do not hold for all summands of $\mathcal{P}$ defined in (E.8) separately, it holds for its initial form $\mathcal{P} = q^2 - (p \cdot q)^2/p^2$. Application of (E.9) is hence justified, and setting $d = 4$ we find

$$I_{\text{ghost}}^{\text{MB}}(p,m) = \frac{3}{2(4\pi)^2}(p^2)^{1-2\kappa-s-t}\frac{\Gamma(s+t+2\kappa+2)\Gamma(2-s-\kappa)\Gamma(2-t-\kappa)}{\Gamma(s+\kappa+1)\Gamma(t+\kappa+1)\Gamma(4-s-t-2\kappa)}. \tag{E.10}$$

Using $\Gamma(z+1) = z\Gamma(z)$ and the result of the momentum integration (E.10), (E.7) becomes

$$\mathcal{D}_{\text{ghost}} = \frac{-3}{128\pi^4}\frac{1}{\Gamma(\kappa)^2}\int_{-i\infty}^{i\infty} ds \int_{-i\infty}^{i\infty} dt \Big(\frac{m^2}{p^2}\Big)^{s+t}(p^2)^{1-2\kappa} \tag{E.11}$$

$$\times\frac{\Gamma(-s)\Gamma(-t)\Gamma(s+t+2\kappa+2)\Gamma(2-s-\kappa)\Gamma(2-t-\kappa)}{(s+\kappa)(t+\kappa)\Gamma(4-s-t-2\kappa)}.$$

The two remaining integrals in (E.11) along the imaginary axis can be evaluated via the residue theorem, closing the integration contour at real positive/negative infinity for $p^2 > m^2/p^2 < m^2$. This step can be automated using the Mathematica packages *MB* [102] and *MBsums* [103]. The result is quoted as

$$\mathcal{D}_{\text{ghost}} = \frac{-3}{128\pi^4}\frac{1}{\Gamma(\kappa)^2}\begin{cases}(m^2)^{1-2\kappa}\mathcal{M}^{\text{IR}}(p,m), & p < 4m,\\ (p^2)^{1-2\kappa}\mathcal{M}^{\text{UV}}(p,m), & \text{else}.\end{cases} \tag{E.12}$$

The functions $\mathcal{M}$ are given by the sums of the residues of (E.11), and explicitly read

$$\mathcal{M}^{\text{IR}} = \mathcal{M}_1^{\text{IR}} + \mathcal{M}_2^{\text{IR}} + \mathcal{M}_3^{\text{IR}}, \tag{E.13}$$

with

$$\mathcal{M}_1^{\text{IR}} = \frac{-1}{24}\Big(\frac{p^2}{m^2}\Big)^2\Gamma(\kappa)\Gamma(\kappa+1)\,_3F_2\Big(\begin{matrix}1,1,1+\kappa\\2,5\end{matrix}\Big|-\frac{p^2}{m^2}\Big), \tag{E.14}$$

$$\mathcal{M}_2^{\text{IR}} = -\sum_{n_1,n_2=0}^{\infty}\frac{(-1)^{n_1+n_2}\Gamma(n_1+n_2+\kappa+1)\big(\frac{m^2}{p^2}\big)^{-n_1}}{n_1!(n_1+2)!n_2!(n_2+2)!(n_1+n_2+1)!\Gamma(n_2+\kappa+1)\Gamma(-n_1-n_2+2-\kappa)}$$

$$\times\Big[(n_2+1)!\Gamma(-n_2-2-\kappa)\Gamma(n_2+\kappa+1)(n_1+n_2)!(n_1+n_2+2)!\Gamma(-n_1-n_2+2-\kappa)$$

$$-(n_2+2)!\Gamma(2-\kappa-n_2)\Gamma(n_2+\kappa)(n_1+n_2+1)!$$

$$\times\Gamma(-n_1-n_2+1-\kappa)\Gamma(n_1+n_2+2\kappa-1)\Big],$$

$$\mathcal{M}_3^{\text{IR}} = \Gamma(\kappa)^2\Big(\frac{1}{2(\kappa-1)}+\frac{1}{6}\frac{p^2}{m^2}\Big[\frac{1}{6}\big(\psi(\kappa)-\log\frac{m^2}{p^2}+\gamma_E\big)-\frac{11}{6}\Big]\Big),$$

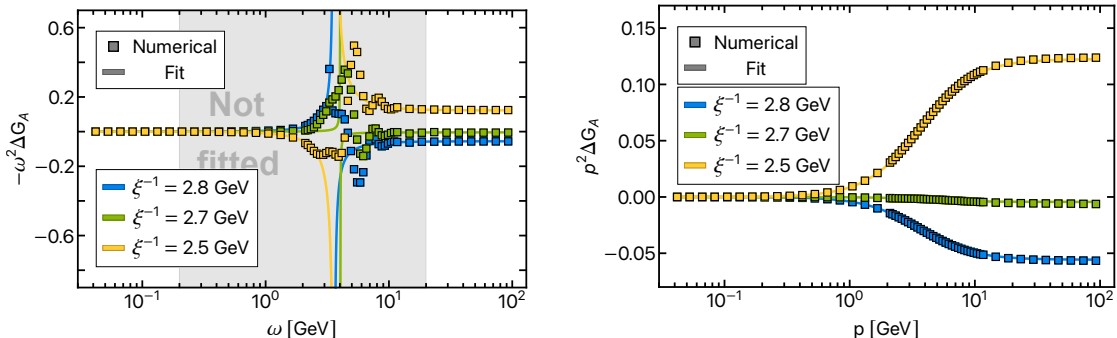

Figure 14: Spectral difference $\Delta G_A$ on the real (left) and imaginary (right) frequency axis. Squares indicate the numerical values of $\Delta G_A$, while solid lines mark the corresponding fit $G_A^{\text{approx}}$ by a pole on the real frequency axis, cf. (37). The best fit for the spectral difference is constructed on the level of the dressing function, as displayed above. The fit describes the Euclidean well. On the Minkowski axis, only the asymptotic tails are fitted, and the fit works well in this regime. In the mid-momentum regime of the real axis however, numerous wiggles suggests that multiple pairs of complex conjugate poles are present. The region is explicitly excluded from the fit.

where $\psi$ is the digamma function and

$$
\mathcal{M}_1^{\text{UV}} = \sum_{n_1,n_2=0}^{\infty} \frac{(-1)^{n_1+n_2}\left(\frac{m^2}{p^2}\right)^{n_1+n_2}\Gamma(-\kappa-n_1+2)\Gamma(-\kappa-n_2+2)\Gamma(2\kappa+n_1+n_2-1)}{(\kappa+n_1)(\kappa+n_2)\Gamma(n_1+1)\Gamma(n_2+1)\Gamma(-2\kappa-n_1-n_2+4)},
$$

$$
\mathcal{M}_2^{\text{UV}} = \sum_{n_1,n_2=0}^{\infty} \frac{(-1)^{n_1+n_2}(n_1+1)!\left(\frac{m^2}{p^2}\right)^{-\kappa+n_1+n_2+2}}{n_1!(n_1+2)!n_2!\Gamma(-\kappa-n_1-n_2+2)\Gamma(\kappa+n_2+1)}
$$

$$
\times \Gamma(\kappa-n_1-2)\Gamma(-\kappa-n_2+2)\Gamma(\kappa+n_2)\Gamma(\kappa+n_1+n_2+1),
$$

$$
\mathcal{M}_3^{\text{UV}} = \sum_{n_1,n_2=0}^{\infty} \frac{(-1)^{n_1+n_2}(n_2+1)!\left(\frac{m^2}{p^2}\right)^{-\kappa+n_1+n_2+2}}{n_1!n_2!(n_2+2)!\Gamma(\kappa+n_1+1)\Gamma(-\kappa-n_1-n_2+2)}
$$

$$
\times \Gamma(-\kappa-n_1+2)\Gamma(\kappa+n_1)\Gamma(\kappa-n_2-2)\Gamma(\kappa+n_1+n_2+1). \tag{E.15}
$$

The double sums appearing in (E.14) and (E.15) can be represented as Kampé de Fériet functions, which generalise the hypergeometric function of two variables to

$$
F_{C:D;D'}^{A:B;B'}\left( \begin{array}{c} a_1,\ldots,a_A : b_1,\ldots,b_B; b'_1,\ldots,b'_{B'} \\ c_1,\ldots,c_C : d_1,\ldots,d_D; d'_1,\ldots,d'_{d'} \end{array} \middle| z_1,z_2 \right) \tag{E.16}
$$

$$
= \sum_{j_1,j_2=0}^{\infty} \frac{(a_1)_{j_1+j_2}\ldots(a_A)_{j_1+j_2}(b_1)_{j_1}\ldots(b_B)_{j_1}(b'_1)_{j_2}\ldots(b'_{B'})_{j_2}}{(c_1)_{j_1+j_2}\ldots(c_C)_{j_1+j_2}(d_1)_{j_1}\ldots(d_D)_{j_1}(d'_1)_{j_2}\ldots(d'_{D'})_{j_2}} \frac{z_1^{j_1}z_2^{j_2}}{j_1!j_2!},
$$

by identifying the respective Pochhammer symbols. Since in numerical implementations special functions such as the Kampé de Fériet function defined in (E.16) are often evaluated via their series representation, we do not reformulate the double sums in (E.14) and (E.15) here. The presented analytic result can be validated by evaluating (E.4) numerically. Note that in particular, with the above expressions at hand, also here we can directly evaluate the diagram at real frequencies $\omega$.

# F  Scale setting and normalisation

In order to provide data which can be compared to the lattice, we need to fix the momentum scale and global normalisation of both fields. This is done by introducing two rescaling factors via

$$Z_{c/A}^{(\text{lat})}(p_{\text{GeV}}) = \mathcal{N}_{c/A} Z_{c/A}(c \cdot p_{\text{internal}}). \tag{F.1}$$

The normalisation of ghost and gluon field $\mathcal{N}_{c/A}$ as well as (common) momentum rescaling factor $c$ are then determined by fitting the Euclidean ghost and gluon dressing functions in (F.1) to the Yang-Mills fRG data of [46], which is itself properly rescaled to match the lattice data of [104]. For the gluon, during the fit an additional constant term $\Delta m_A^2$ needs to be allowed on the level of $\Gamma_{AA}^{(2)}$ in order to compensate for possible differences in the constant part of $\Gamma_{AA}^{(2)}$. Note that this term is only introduced to correctly determine $\mathcal{N}_A$ and $c$, and is removed again after rescaling.

# G  Spectral difference

This appendix discusses the fitting procedure that is used to take the spectral difference (35), i.e. the difference between the spectral and full gluon propagator, into account. The spectral difference is evaluated on both real and imaginary frequency axis. The obtained result is fit with a simple pole on the real frequency axis, comp. (37), using Mathematica's NonlinearModelFit routine. We select Newton's method for the optimisation and explicitly specify gradient and hessian of the fit function. The trust region method is employed for step control. Further, we assign weights $w_i$ to the (real and imaginary) frequency grid points. It turned out to be beneficial for the convergence to choose $w_i = |p_0^2|$, where $p_0$ stands for both real or imaginary frequencies. This results simply in fitting the spectral difference of the gluon dressing function instead of the propagator. In consequence, the fitting routine puts a lot more weight on the UV instead of the IR, which is the case when fitting the propagator. The fact that this increases

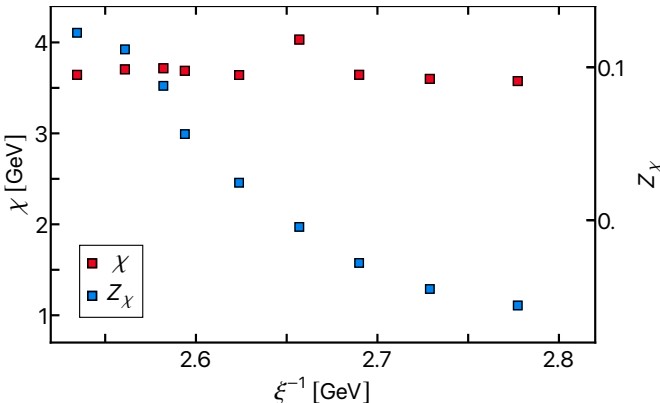

Figure 15: Evolution of the pole positions (red) and residues (blue) of the spectral difference fit $G_A^\chi$ defined in (37) under change of $m_A^2$. The residues mirror the spectral violations $\mathcal{V}_{\text{spec}}$ defined in (39) and shown in Figure 8. If the spectral violation is negative, i.e. the Källén-Lehmann part $G_A^\chi$ is smaller than the full gluon propagator $G_A$, the spectral difference and, correspondingly the residue, are positive, and vice versa. Decreasing stability of the iteration and worsening fit precisions towards smaller $m_A^2$, which can be also seen when comparing the spectral difference $\Delta G_A$ and its fit $G_A^\chi$, shown in Figure 14, also manifest themselves in non-monotonous behaviour of $\chi$.

stability can be well understood considering that the effect of the deep IR behaviour of the gluon propagators in the DSE diagrams (comp. Figure 2) is relatively subleading compared to the UV behaviour.

In order to assess the quality of the employed fit, it is hence sensible to consider the spectral difference for the gluon dressing function, as shown in Figure 14. The left panel shows the spectral difference as compared to the fit on the real-time axis. The grey shaded area is explicitly excluded from the fit, such that only the asymptotic tails are taken into account. As emphasised in Section 5.1, it can be clearly seen that the employed fit function is not able to capture the full structure of the spectral difference. The asymptotic tails are well fit, however. Taking a closer look at the excluded region in fact suggest the existence of multiple pairs of complex conjugate poles. A single complex conjugate pole term accounts for each one local maximum and minimum in the spectral difference. A rough estimate for the number of (leading order) pairs of poles can thus be obtained by just counting positive/negative peaks.

The Euclidean spectral difference for the gluon dressing is displayed in the left panel of Figure 14. As for the asymptotic regions on the real frequency axis, the fit works well here. However, comparing the fit quality between the different values of $m_A^2$, the worst fit is obtained for smallest $m_A^2$.

# H   Numerical procedure

This appendix elaborates on the numerical treatment of the spectral integrals as well as the spectral integrands.

## H.1   Spectral integration

The spectral integrals of the form

$$\int_{\{\lambda_i\}} \prod_i \lambda_i \rho_i(\lambda_i) I_{\text{ren}}(p, \{\lambda_i\}), \tag{H.1}$$

where $I_{\text{ren}}$ is the renormalised spectral integrand (comp. (29) or (31)), are evaluated numerically on a logarithmic momentum grid of 100-200 grid points with boundary $(p_{\min}, p_{\max}) = (10^{-4}, 10^2)$, identically for the Euclidean and Minkowski axis. We use a global adaptive integration strategy with default multidimensional symmetric cubature integration rule. After spectral integration, the diagram is interpolated with splines in the Euclidean and Hermite polynomials in the Minkowski domain, both of order 3. The spectral function is then computed from the interpolants. Note that, a priori, due to (8), the domain of the ghost spectral function is given by the momentum grid. The integration domain of the spectral integral of the ghost spectral parameter has to be bounded by $(p_{\min}, p_{\max})$, in order to not rely on the extrapolation of the spectral function beyond the grid points. Due to numerical oscillations at the very low end of the grid, we choose $(\lambda_c^{\min}, \lambda_c^{\max}) = (10^{-3.5}, 10^2)$. Convergence of the integration result with respect to increase of the integration domain has been explicitly checked.

## H.2   Spectral integrands

The numerical performance of the spectral integrations presented in Appendix B is sped-up by up to two orders of magnitude by using interpolating functions of the numerical data. The interpolants are constructed by first discretising the integrand inside the three-dimensional $(p, \lambda_1, \lambda_2)$ cuboid defined by $p \in 10^{\{-4,2\}}$, $\lambda_{1,2} \in 10^{\{-4,4\}}$. As for the momentum grid for the

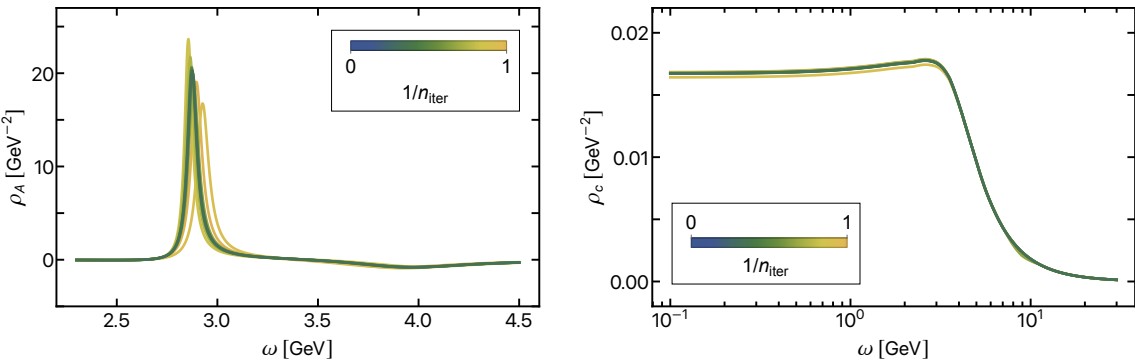

Figure 16: Convergence of a gluon (left) and ghost (right) spectral function through iteration of the DSE for $m_A^2 = -1.1$. As an initial guess, the solution for $m_A^2 = -1$ is used. The colour coding indicates the iteration number $n_{\text{iter}}$. After about 10 iterations, the curves become visually indistinguishable, i.e. the iteration converges.

spectral integration, we use the same cuboid for the real- and imaginary-time domain. We use 60 grid points in the momentum and 160 grid points in the spectral parameter integration, both with logarithmic grid spacing. For the real-time expressions, we divide into real and imaginary part of the integrands. Both real and imaginary parts of the discretised Minkowski as well as the Euclidean expressions are then interpolated by three-dimensional splines inside the cuboid. The resulting interpolating functions are then used in the spectral integration.

### H.3 Convergence of iterative solution

The iterative procedure applied to solve the coupled system of DSEs in this work is described in Section 3.5. For each value of $m_A^2$, the iterations is initiated with spectral functions from the previous (larger) value of $m_A^2$. It converges rapidly, see Figure 16. The very first initial guess for the gluon spectral function has been obtained heuristically by trial-and-error from previous iteration results and has not been stored. For the ghost spectral function, a massless pole in the origin with residue 1 was used.

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
