# Peer review of "On the complex structure of Yang-Mills theory"

_SciPost Physics, doi:SciPost Phys. Core 8, 048 (2025)_

## Round 4 · Referee Report · Anonymous (Referee 1) · 2024-8-2

Report

It is a pity that the authors did not make any real change in the last revision: they just added an other Appendix where they basically admit that they are using CF model as an approximation for YM theory. It is a relevant point which should be discussed from the beginning. Rather, it seems that they are hiding the discussion within an other appendix (and there are too many! It was suggested to reduce their number!)

Since the author refuse to meet the requirements of the referees, I must suggest that the paper is not published.

Recommendation

Reject

---

## Round 4 · Referee Report · Anonymous (Referee 2) · 2024-9-2

Report

I am satisfied with the changes made in the draft and therefore recommend publication.

Recommendation

Publish (surpasses expectations and criteria for this Journal; among top 10%)

---

## Round 4 · Author Response

In order to address the repeated criticism regarding the differentiation between Yang-Mills theory and the Curci-Ferrari model, we added another Appendix detailed the difference between the two and how we interpret our results.
We hope that this finally clarifies any potential issues a reader might have with distinguishing the two.

---

## Round 4 · List of Changes

• New appendix detailing the difference between YM-Theory and the Curci-Ferrari model
  • Referenced the new appendix in the main text

---

## Editorial Decision

published